# In eubacteria, unlike eukaryotes, there is no evidence for selection favouring fail-safe 3' additional stop codons

Alexander T. Ho*, Laurence D. Hurst

Milner Centre for Evolution, University of Bath, Bath, United Kingdom

* a.t.ho@bath.ac.uk

## Abstract

Errors throughout gene expression are likely deleterious, hence genomes are under selection to ameliorate their consequences. Additional stop codons (ASCs) are in-frame nonsense 'codons' downstream of the primary stop which may be read by translational machinery should the primary stop have been accidentally read through. Prior evidence in several eukaryotes suggests that ASCs are selected to prevent potentially-deleterious consequences of read-through. We extend this evidence showing that enrichment of ASCs is common but not universal for single cell eukaryotes. By contrast, there is limited evidence as to whether the same is true in other taxa. Here, we provide the first systematic test of the hypothesis that ASCs act as a fail-safe mechanism in eubacteria, a group with high read-through rates. Contra to the predictions of the hypothesis we find: there is paucity, not enrichment, of ASCs downstream; substitutions that degrade stops are more frequent in-frame than out-of-frame in 3' sequence; highly expressed genes are no more likely to have ASCs than lowly expressed genes; usage of the leakiest primary stop (TGA) in highly expressed genes does not predict ASC enrichment even compared to usage of non-leaky stops (TAA) in lowly expressed genes, beyond downstream codon +1. Any effect at the codon immediately proximal to the primary stop can be accounted for by a preference for a T/U residue immediately following the stop, although if anything, TT- and TC- starting codons are preferred. We conclude that there is no compelling evidence for ASC selection in eubacteria. This presents an unusual case in which the same error could be solved by the same mechanism in eukaryotes and prokaryotes but is not. We discuss two possible explanations: that, owing to the absence of nonsense mediated decay, bacteria may solve read-through via gene truncation and in eukaryotes certain prion states cause raised read-through rates.

## Author summary

In all organisms, gene expression is error-prone. One such error, translational read-through, occurs where the primary stop codon of an expressed gene is missed by the translational machinery. Failure to terminate is likely to be costly, hence genomes are

---

**Data Availability Statement:** Scripts can be found at https://github.com/ath32/ASCs. All other relevant data are within the manuscript and its Supporting Information files.

**Funding:** This work was supported by the European Research Council (Grant EvoGenMed ERC-2014-ADG 669207 to L.D.H). For more information regarding ERC activities, please visit https://erc.europa.eu/. The funders had no role in study design, data collection and analysis, decision to publish, or preparation of the manuscript.

**Competing interests:** The authors have declared that no competing interests exist.

under selection to prevent this from happening. One proposed error-proofing strategy involves in-frame proximal additional stop codons (ASCs) which may act as a 'fail-safe' mechanism by providing another opportunity for translation to terminate. There is evidence for ASC enrichment in several eukaryotes. We extend this evidence showing it to be common but not universal in single celled eukaryotes. However, the situation in bacteria is poorly understood, despite bacteria having high read-through rates. Here, we test the fail-safe hypothesis within a broad range of bacteria. To our surprise, we find that not only are ASCs not enriched, but they may even be selected against. This provides evidence for an unusual circumstance where eukaryotes and prokaryotes could solve the same problem the same way but don't. What are we to make of this? We suggest that if read-through is the problem, ASCs are not necessarily the expected solution. Owing to the absence of nonsense-mediated decay, a process that makes gene truncation in eukaryotes less viable, we propose bacteria may rescue a leaky stop by mutation that creates a new stop upstream. Alternatively, raised read-through rates in some particular conditions in eukaryotes might explain the difference.

## Introduction

Errors throughout transcription, translation, and post-translational modification can, and do, happen all the time [1–5]. Whilst an invaluable source of novelty that drives evolution [6], the majority of these errors are likely deleterious [1–3, 6–8]. Genomes may therefore be under selection to mitigate their consequences. This has been supported by bioinformatic studies of stop codon usage in gene locations other than that of the canonical stop. For example, it has been suggested that adenine enrichment at the fourth coding sequence residue in bacterial genes may promote translation termination following a frameshift event at the initiating ATG that allows an out-of-frame stop codon to be read [9, 10]. In 5' leading regions, in-frame stop codons are enriched and postulated to rapidly terminate premature translations [11] (i.e. those that occur before the ribosome reaches the recognised start codon of the mRNA). Selection on the primary stop codon is also thought to be error related [12–14]. Experimental evidence from bacterial studies suggest the three stops differ in their read-through rates [14–21]. Notably the least leaky of the three, TAA, is the preferred codon, especially in the most highly expressed genes [13]. In this study, we consider the hypothesis that additional stop codons (ASCs) occur after the primary stop codon as a fail-safe mechanism to minimise the costs of stop codon readthrough [22]. This question is important both as a means to address the importance of error-proofing to genome evolution but potentially also for optimal transgene design.

Although ribosomes normally terminate translation at stop codons there is a chance that an ectopic amino acid is inserted, allowing translation to continue in the same frame for the generation of extended polypeptides [23, 24]. The primary cause appears to be aberrant recognition by near-cognate tRNAs [25, 26] or other tRNA species [27]. While read-through rates vary depending both on the stop codon and local sequence context [28], read-through rates are typically orders of magnitude higher than the mutation rate [29–33], rendering read-through a potentially significant fitness-modifying trait. While there may be beneficial consequences, such as increased proteome diversity [34], the best evidence suggests that it is largely non-adaptive [8]. Selection for the least leaky stop in highly expressed genes [13] provides strong support for the notion that selection acts to reduce read-through rates as it is most commonly a deleterious error. Possible costs include energetic wastage owing to unnecessary

translation [35] and creation of potentially toxic or sticky novel peptides. Resource wastage can be acute if the ribosome needs to be recovered, as can happen, for example, if it moves into a polyA tail as both RNA and protein can be targeted for destruction [36–38]. In theory, the presence of ASCs downstream may alleviate some of these costs by reducing the amount of additional amino acids added to erroneous polypeptide chains [39] and preventing polyA associated destruction. We herein refer to such a system as the 'fail-safe' hypothesis.

The fail-safe hypothesis has been most thoroughly examined in eukaryotes, notably in yeast [39], and two ciliate species which have reassigned their genetic code such that TGA is the only stop codon [40]. In yeasts, a statistical excess of UAA at the third codon downstream of TAA-terminating genes points towards a maintenance of ASCs by selection in a manner dependent on expression level [39]. This was corroborated in ciliates, where ASCs appear downstream of the primary stop more often than expected by chance given the base composition of 3' regions [40]. Given that the excess is larger in ciliates than in yeast, it was proposed that ASCs are under variable selection intensity dependent on readthrough rate, which in turn may vary between species [40]. This, however, remains *post hoc* speculation.

In bacteria tests of the fail-safe hypothesis are lacking. One study found tandem ASCs (those which immediately follow the primary stop) are over-represented, being seen in 7% of *E. coli* genes [41]. However, the experimentally estimated termination efficiency of tandem stops were below the expected rate and it was postulated that *prima facie* over-representation in the genome could be attributed instead to the preference for a tetranucleotide containing +4U, thought to improve the termination efficiency of the primary stop [41–45]. +4U in this context refers to the base immediately after the primary stop. A +4U base biases the first codon after the primary stop towards a second stop codon as all stops start T/U.

More recently, one study has widened the investigation to ASCs in the following 5 in-frame codon positions. Such ASCs are reported in 8% of *E. coli* genes [13], however, although this figure concords with the findings of Major and colleagues [41], the authors do not comment on whether this is higher, lower or the same as expected given more codon positions are being considered. More generally, it is unknown whether ASC frequency downstream is higher than expected under a GC-controlled null in any eubacteria. Preliminary data weakly argue against the fail-safe hypothesis as there is no preference for UAA, UGA or UAG as an ASC downstream of the primary stop [13]. While, however, one might imagine selection that favours ASCs might also be strong enough to bias usage towards the strongest stop (UAA), this is a second order effect compared with selection for any ASC in leaky genes.

Differential leakiness of stop codons in eubacteria provides a foundation for testing the fail-safe hypothesis. While UAA [29], UGA [30, 31], and UAG [29, 32, 33] are all subject to readthrough, they do so to differing degrees. The mechanistic basis for this variation is thought to relate to the specificity and abundance of release factors. The stop codons are recognised by a class I release factor [46–49], with their dissociation mediated by class II release factors following peptide release [50]. In bacterial lineages decoded according to translation table 11 (TT11), the class I release factors responsible are RF1 and RF2. UAG is recognised by RF1, UGA is recognised by RF2, and UAA is recognised by both RF1 and RF2 [48, 51, 52]. It is thought that the ability of UAA to bind both RF1 and RF2 contributes to it being the least 'leaky' stop. No matter what the mechanism, the selection of ASCs is likely to be highest in UGA-terminating genes and weakest for UAA, all else being equal.

In addition to termination efficiency, there are at least two other predictors of stop codon usage, GC pressure and expression level, when comparing across genes and genomes. While between genomes genomic GC is a strong predictor of UAA and UGA alone, UAG and UGA, with identical nucleotide contents, show dissimilar trends, UGA usage being positively correlated with genomic GC while UAG usage is uncorrelated [13, 14, 53]. This is conjectured to

relate to co-evolution between RF1:RF2 ratios and GC content [14]. Within genomes it is considered that highly expressed genes should be under selection to employ UAA this being the least leaky. Indeed, while across bacteria UAA usage is well predicted by GC pressure, it is found to be enriched in highly expressed genes (HEGs) even in GC rich genomes [13, 14]. The resistance to GC pressure in HEGs is consistent with the notion that the net effect of readthrough is a combined function of the per translation leakage rate and the number of translation events any given transcript is subject to.

Here we provide the first systematic test of the fail-safe hypothesis applied to eubacteria. We interrogate the 3' UTRs of a large sample of phylogenetically relatively independent bacterial species for enrichment of ASCs. In acknowledgment of prior studies, we control for GC pressure [13, 14, 53]. We ask whether we can detect ASCs at rates higher than expected given underlying nucleotide content, and whether 3' UTR codon switches seen in closely related species are biased towards ASC deposition compared to null (determined by out of frame rates). Further, we ask whether highly expressed genes have more ASCs and whether expression level and primary stop usage predicts ASC usage. The most extreme difference should be between highly expressed TGA ending genes, which should have strong ASC selection, and lowly expressed TAA ending genes in which fail-safe selection should be the weakest. We also ask if the presence of an ASC predicts the downstream presence of further ASCs and whether mollicutes employing only two stops under-employ the codon that isn't a stop.

The tests are, however, complicated by the fact that stop codon efficiency is also dictated by local genomic context [28]. Indeed, it has been observed that nucleotide substitution rate increases with downstream distance from the stop codon with no obvious plateau within the next six downstream 'codons' [12], bringing attention to this region as a potential influencer of termination efficiency. Such regions may be directly involved in the formation of termination complexes that include the ribosome [45]. As noted, one downstream element thought to affect termination is the nucleotide at position +4 [41, 42, 54, 55]. In eukaryotes, +4C is associated with an increase to ca. 3% readthrough in certain genomic contexts [55], whereas +4U is highly preserved in all three domains of life and thought to reduce readthrough rate via improved cross-linking with RF2 [42]. This is problematic as it tends to increase the frequency of 3' in-frame stops at the first downstream codon compared to the simplest null model. At a greater scale, at least a hexanucleotide sequence may affect termination efficiency [44, 55, 56]. Whilst this evidence was found in eukaryotes, it cannot be discounted that the local genomic context affecting readthrough rates in bacteria could extend beyond the fourth site nucleotide. Thus, we attempt to control for downstream motif preferences, in addition to GC content, in our assessment of whether ASCs are selected for error-control. We find that, in contrast to eukaryotes, the great majority of our evidence argues against the notion that 3' ASCs are selectively favoured. We speculate as to why this might be.

## Results

### Nucleotide controlled simulations suggest genome-wide avoidance, not enrichment, of ASCs

A prediction of the fail-safe 3' stop hypothesis is that stop codons should be enriched immediately after the primary stop. Thus, we assessed genomes for ASC enrichment through comparison against a null model where downstream 3' codons are chosen according to dinucleotide content only. This was achieved by the simulation of 10,000 dinucleotide-controlled 3' UTRs per genome, the calculated mean ASC frequencies being the 'expected' value and the Z-score being the deviation from this mean normalised to the standard deviation of the simulations. A positive Z-score is an instance where ASCs are overused compared to null.

The null neutral expectation was that there is no difference between the ASC frequencies of the real genomes and simulated sequences hence 50:50 split of positive and negative Z-scores. We instead find there to be significant variation from this ratio when considering the UTR as a whole but, unexpectedly, with an excess of instances of under-usage of stops (from codon position +1 to +6; 13/644 Z > 0, p < 2.2 x $10^{-16}$, two-tailed binomial test). The same under usage is seen at all sites when considered individually (p < 2.2 x $10^{-16}$ for all positions, two-tailed binomial tests; 89/644 Z > 0 at position +1, 56/644 Z > 0 at position +2, 36/644 Z > 0 at position +3, 35/644 Z > 0 at position +4, 48/644 Z > 0 at position +5, 40/644 Z > 0 at position +6). All significant findings survive multi-test correction (p < 0.05/6).

These results accord with what we see if we consider the proportion of genomes showing significant deviation compared to null (|Z| > 1.96). In this instance, the null expectation of the binomial test is no longer 50:50, rather that 95% of genomes will not be significantly deviated and 5% will. There is a significant variation from this ratio when considering UTR *en mass* (p < 2.2 x $10^{-16}$ for the whole UTR (553/644 genomes) and at each position (p < 2.2 x $10^{-16}$ at position +1 (177/644 genomes), p < 2.2 x $10^{-16}$ at position +2 (129/644), p < 2.2 x $10^{-16}$ at position +3 (136/644);p < 2.2 x $10^{-16}$ at position +4 (113/644), p < 2.2 x $10^{-16}$ at position +5 (92/644), p = 3.1 x $10^{-12}$ at position +6 (77/644), two-tailed binomial tests), all surviving multi-test correction (p < 0.05/6). Closer examination again indicates that significant enrichment (one-tailed test, therefore we now use Z > 1.64) occurs less than expected by chance (p < 1.6 x $10^{-13}$ for the whole UTR (1/644), p = 2.8 x $10^{-10}$ at position +1 (4/644), p = 1.6 x $10^{-13}$ at position +2 (1/644), p = 4.5 x $10^{-15}$ at position +3 (0/644), p = 4.5 x $10^{-15}$ at position +4 (0/644), p = 1.6 x $10^{-13}$ at position +5 (1/644), p = 4.5 x $10^{-15}$ at position +6 (0/644), one-tailed binomial tests). Indeed, when we consider under-enrichment (Z < -1.64), we find more significant results than expected by chance (p < 2.2 x $10^{-16}$ for whole UTR (570/644), p < 2.2 x $10^{-16}$ at position +1 (230/644), p < 2.2 x $10^{-16}$ at position +2 (206/644), p < 2.2 x $10^{-16}$ at position +3 (204/644), p < 2.2 x $10^{-16}$ at position +4 (176/644), p < 2.2 x $10^{-16}$ at position +5 (135/644), p = 2.9 x $10^{-12}$ at position +6 (119/644), one-tailed binomial tests). These results provide no *prima facie* support for the fail-safe hypothesis and, if anything, argue for ASC avoidance.

Is there anything peculiar about the genomes for which we find under usage of ASCs? As all three stop codon variants are AT-rich by nature, they are more likely to appear in AT-rich genomes by chance. The fail-safe hypothesis therefore predicts selection to retain ASCs most strongly in GC-rich genomes, where a dearth of ASCs is expected in the absence of selection. Our results are contra to this prediction, as we find a significant negative correlation between Z-score and GC3 content (p < 2.2 x $10^{-16}$, ρ = -0.64, Spearman's rank correlation) (Fig 1). This trend is consistent at all positions +1 to +6 (S1 Fig) with the magnitude of the gradient decreasing with 3' distance (S2 Fig). This result is repeated when considering raw ASC frequency instead of Z-score (S3 Fig). Indeed, it appears that it is where ASCs are predicted to be most needed that they most under-employed.

### 3' codon switches from stop to non-stop are more common in-frame than out-of-frame, suggesting avoidance of ASCs

Above, we not only find no evidence for ASC enrichment but for ASC avoidance. Could this be because genomes specifically remove ASCs at a higher rate than chance? Alternatively, perhaps switches from non-stop to stop occur at a lower rate than expected. We investigate both of these possibilities through analysing codon switches from stop to non-stop, and vice versa, in orthologous gene triplets. We employ 29 sets of triplet species (a paired ingroup and an outgroup) and consider the results *en mass*. For null expectations we employ the comparable rate (stop->non-stop, non-stop->stop) in the +1 reading frame of the 3' domain.

**All positions (+1 to +6)**

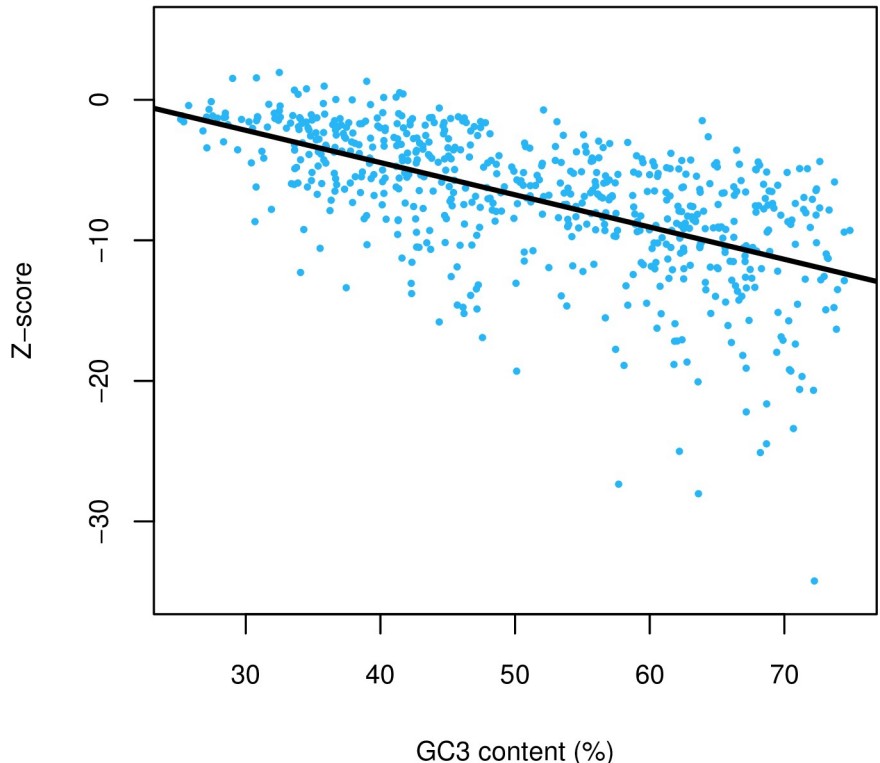

**Fig 1. Z-scores, measuring deviation in ASC frequency from a null model (10,000 simulations), plotted against the genomic GC3 content of filtered TT11 bacterial genomes.** A significant negative relationship is observed between Z-score and genomic GC3 across positions +1 to +6 (p < 2.2 x $10^{-16}$, ρ = -0.64, Spearman's rank correlation).

Considering all codons (Table 1) regardless of position in our dataset of the orthologous genes, we find the frequency of in-frame codon switches from non-stop to stop in 3' UTR codons to be no different to the same switch in out-of-frame codons of the same sequences (p = 0.31, $\chi^2$ = 1.0, Chi$^2$ test). Consistent with selection to avoid ASCs, switches from stop to non-stop occur significantly more often in-frame than out-of-frame (p = 0.0024, $\chi^2$ = 9.2, Chi$^2$ test). Hence not only are in-frame stops not deposited in 3' UTRs more than chance, they are if anything avoided. Both of these results corroborate the findings of our initial binomial tests and argue strongly against the fail-safe hypothesis in bacteria.

Considering each position individually tells a similar story. For the vast majority of codon switches at each position, there is no difference in switch rate between in-frame and out-of-frame codons (S1 Table). Exceptions to this are found at position +4, where switches from stop to non-stop are significantly more common in-frame than out-of-frame, and at position +5,

**Table 1. Codon switch (from stop to non-stop and non-stop to stop) counts and frequencies compared between the in-frame and out-of-frame 3' UTR codons of 29 triplets of closely related bacterial genomes.**

| Switch | In-frame | | | +1 Frame-shift | | | Chi$^2$ p-val |
|---|---|---|---|---|---|---|---|
| | Ancestral codons | Switch count | Switch frequency | Ancestral codons | Switch count | Switch frequency | |
| Stop > NS | 2,693 | 655 | 0.243 | 3,146 | 679 | 0.216 | 0.0024 |
| NS > Stop | 51,525 | 664 | 0.0129 | 52,372 | 704 | 0.0134 | 0.31 |

where switches from non-stop to stop are significantly less common in-frame than out-of-frame. Both results are consistent with rejection of the fail-safe hypothesis, however do not survive even generous Bonferroni correction (p > 0.05/6). At position +1, switches from non-stop to stop are significantly more common in-frame than out-of-frame, though this is likely explained by selection for +4T.

## No enrichment of ASCs in highly expressed genes

The above tests provide no support for the fail-safe hypothesis but consider genes equally, regardless of the primary stop codon and expression level. Selection for termination efficiency is thought to be highest in HEGs [13, 57] under the assumption that the net effect of readthrough is a function of the number of translation events the transcripts of any given gene are subject to. If the fail-safe hypothesis of ASCs is true, we therefore expect ASC frequencies to be significantly higher in HEGs than LEGs. This, however, does not seem to be the case. Unlike what is seen in yeast [39], there were no significant differences between the ASC frequencies of HEGs and LEGs at any position even before multi-test correction (p = 0.95 for whole UTR, p > 0.05 for all positions, Wilcoxon signed-rank tests; S4 Fig), suggesting that either expression level has no influence over the negative effects of readthrough or ASCs do not significantly affect the ability of a transcript to avoid these consequences. This test is however limited by small genome sample size. Through manually adding enrichment of stop codons to our data we find that a ~35% increase in HEGs compared to LEGs is required to retrieve a signal. Hence, we can be confident that ASC frequencies in our HEGs dataset do not exceed those seen in LEGs by this margin. We cannot investigate codon switches in highly and lowly expressed gene groups, as the PaxDb database does not contain compatible data to match the ATGC data we used for this analysis.

## No enrichment of ASCs in TGA-terminating HEGs compared to TAA-terminating LEGs

The HEG/LEG analysis, whilst also negative, does not allow for covariance between expression level and usage of different stop codons. Notably the least leaky stop (TAA) is also the preferred one in the highly expressed genes [13, 14], which has the potential to dampen any differences between HEGS and LEGs. Under the fail-safe hypothesis, we expect TGA-terminating HEGs (high readthrough, high expression) to have the strongest selection for ASCs and TAA-terminating LEGS (low readthrough, low expression) to have the weakest. However, we find no significant difference between these groups when considering the whole UTR (p = 0.36, Wilcoxon signed-rank test). Aside from position +1, there is no significant difference between TGA-terminating HEGs and TAA-terminating LEGs at a single position scale (p = 0.060 for position +2, p = 1 for position +3, p = 0.83 for position +4, p = 0.60 for position +5, p = 0.62 for position +6, Wilcoxon signed-rank tests). Even at position +1 the enrichment of ASC in the TAG/HEG class is a barely significant trend (p = 0.041, Wilcoxon signed-rank test) that does not survive Bonferroni correction (p > 0.05/6) (Fig 2). We thus find no evidence to support the notion of ASC selection, apart from a possible very weak effect at position +1.

## ASC enrichment at position +1 is peculiar to TGA-terminating genes

If the above exceptional result at position +1 is owing to selection we might also expect the enrichment to be seen in other TAG and TAA highly expressed expressed genes. If, alternatively, it is a motif preference associated with TGA, we might expect it to be seen in lowly expressed TGA terminating genes but not necessarily elsewhere.

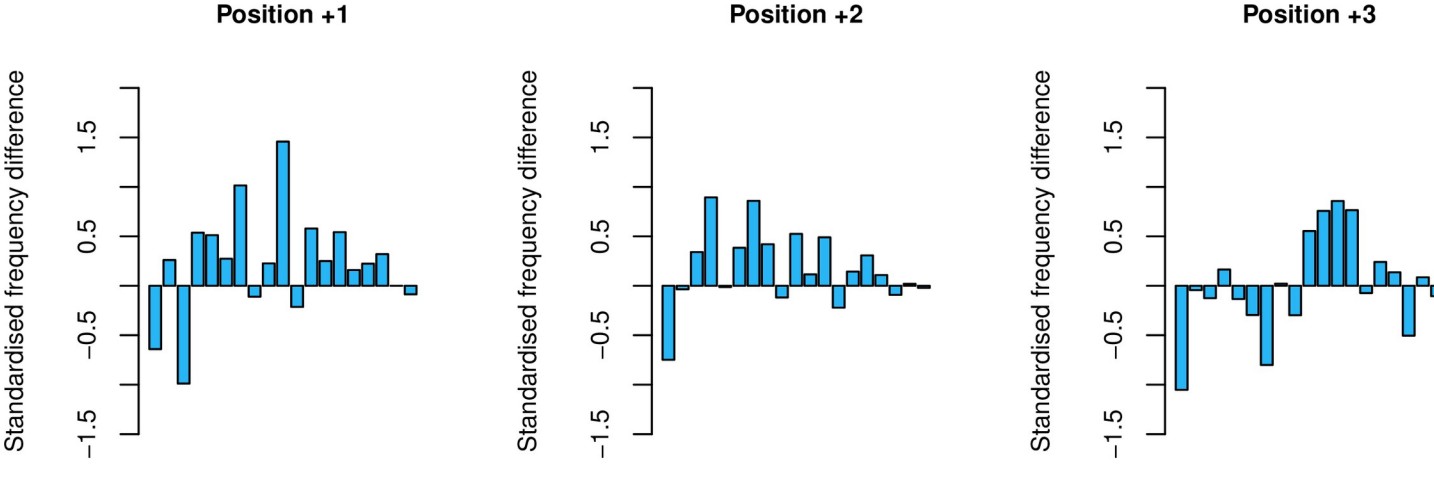

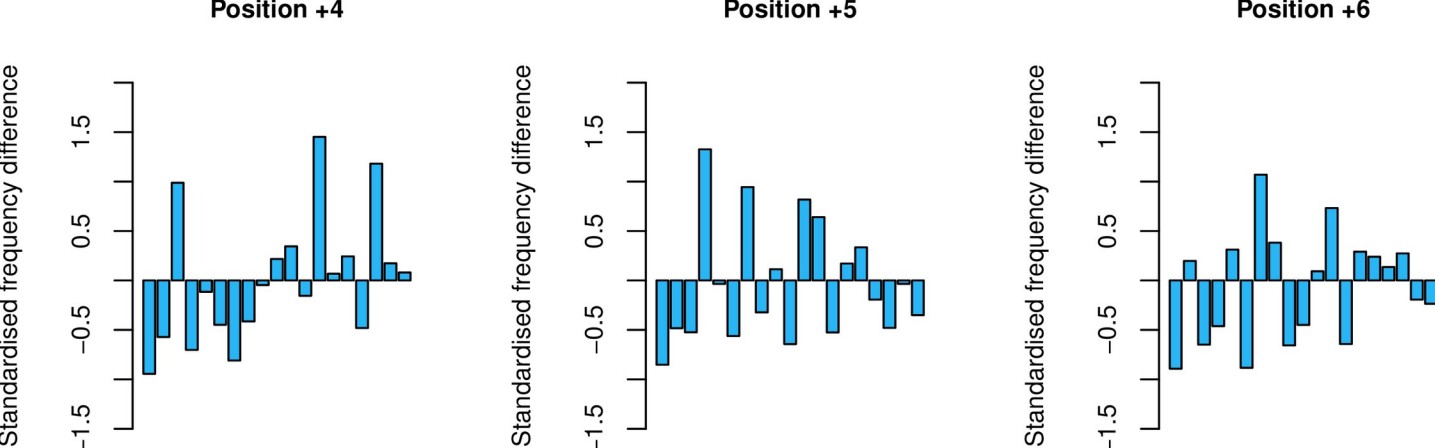

**Fig 2. ASC frequencies of TGA-terminating HEGs compared to TAA-terminating LEGs.** Each bar represents one genome, with bar heights representative of the standardised difference between the two groups. There is no significant difference between TGA-terminating HEGs and TAA-terminating LEGs at positions +2 to +6 (p = 0.060 for position +2, p = 1 for position +3, p = 0.83 for position +4, p = 0.60 for position +5, p = 0.62 for position +6, Wilcoxon signed-rank tests). There is prima facie significant difference between TGA-terminating HEGs and TAA-terminating LEGs at position +1 (p = 0.041, Wilcoxon signed-rank test), but this does not survive Bonferroni correction.

To examine these possibilities we consider all combinations of expression level and primary stops in the assessment of ASC frequency (Fig 3). Considering the whole UTR (+1 to +6) we find evidence for heterogeneity when considering all genes regardless of expression level (p = 0.01, $\chi$ = 8.79, Kruskal-Wallis). However, if we remove position +1 from this analysis, significant heterogeneity cannot be recovered (p = 0.57, $\chi$ = 1.12, Kruskal-Wallis). Indeed, we find that ASC enrichment is particular to position +1 and a peculiarity of TGA terminating genes weakly seen at all expression levels. We established this by first testing for heterogeneity between ASC usage dependent on the primary stop at position +1. When considering all genes (p = 1.9 x $10^{-15}$, $\chi$ = 67.81, Kruskal-Wallis) and LEGs (p = 0.032, $\chi$ = 6.91, Kruskal-Wallis) we see evidence for such heterogeneity. For HEGs ASC usage is highest for TGA terminating

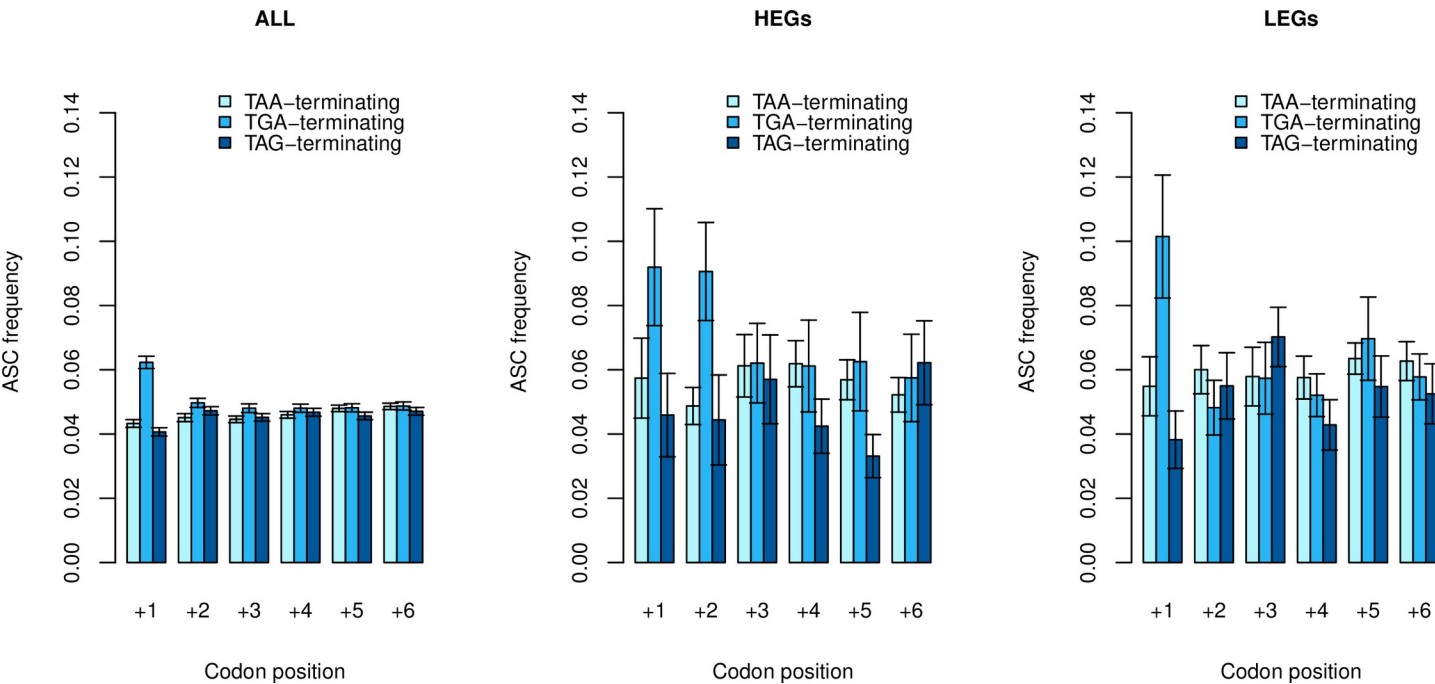

**Fig 3. ASC frequencies calculated in TAA, TGA and TAG-terminating genes for all genes, highly expressed genes (HEGs) and lowly expressed genes (LEGs).** Error bars represent bootstrapped standard error. We find significant differences between primary stop groups at position +1 when considering all genes (p = 1.89 x $10^{-15}$, $\chi$ = 67.81, Kruskal-Wallis) and LEGs (p = 0.032, $\chi$ = 6.91, Kruskal-Wallis), but not HEGs (p = 0.14, $\chi$ = 3.97, Kruskal-Wallis). We instead observe significant enrichment at position +2 in HEGs (p = 0.029, $\chi$ = 7.09, Kruskal-Wallis). Signals at position +1 in LEGs and at position +2 in HEGs do not survive Bonferroni correction. For all other positions, there was no significant difference in any expression group (p > 0.05, Kruskal-Wallis).

genes but not significantly so (p = 0.14, $\chi$ = 3.97, Kruskal-Wallis). Similarly the significance at position +1 in LEGs does not survive Bonferroni correction (p < 0.05/6). With some evidence for heterogeneity, we proceed to post-hoc Wilcoxon signed-rank tests for the two significant cases these indicating in each case, enrichment is highest in TGA-terminating genes (position +1 all genes: TGA > TAA, p < 2.2 x $10^{-16}$; TGA > TAG, p < 2.2 x $10^{-16}$; position +1 LEGs: TGA > TAA, p = 3.0 x $10^{-3}$; TGA > TAG, p = 1.3 x $10^{-3}$, Wilcoxon signed-rank tests). That we do not find significant deviation between primary stops at position +1 in HEGs is surprising, however likely comes as a direct consequence of small sample size.

Confirming the lack of signal outside of position 1, for all such positions, there is no significant difference in any expression group (p > 0.05, Kruskal-Wallis), with one exception, this being significant enrichment at position +2 in HEGs (p = 0.029, $\chi$ = 7.09, Kruskal-Wallis). Here too the effect is most pronounced for TGA terminating genes (position +2 HEGs: TGA > TAA, p = 6.9 x $10^{-3}$; TGA > TAG, p = 0.027, Wilcoxon signed-rank tests), but neither the original test nor the subtest survive Bonferonni correction.

## +4T nucleotide preference rather than ASC selection best explains ASC enrichment at codon 1

Above we have shown that TGA-terminating genes are commonly immediately followed by ASCs. There are two hypotheses for this: (i) a general enrichment of thymine at the fourth coding residue that enables more effective termination [13, 41, 42], most especially true for TGA due to its unique recognition by RF2 alone, and (ii) an enrichment of ASCs in response to TGA leakiness. Several lines of evidence argue in favour of the former.

First, we sought to establish whether there was general +4T enrichment. To this end we calculated the frequency of T-starting codons at position +1 and compared it to the average T-starting codon frequency from positions +1 to +6. T-starting codons at position +1 were found to be enriched compared to other downstream positions ($p < 2.2 \times 10^{-16}$, Wilcoxon signed-rank test). However, this is not necessarily attributable to the presence of position +1 ASCs. In repeating the same methodology, we find the frequency of all non-stop T-starting codons to be significantly enriched at position +1 compared to the UTR average in genes that don't have a position +1 ASC ($p < 2.2 \times 10^{-16}$, Wilcoxon signed-rank test). This effect is most heavily influenced by TGA-terminating genes, in which T-starting non-stop codons are more enriched at position +1 compared to the UTR average ($p < 2.2 \times 10^{-16}$, Wilcoxon signed-rank test) than seen in TAA-terminating genes ($p = 4.7 \times 10^{-14}$, Wilcoxon signed-rank test) and TAG-terminating ($p = 0.9951$, Wilcoxon signed-rank test) genes.

Second, we find ASC frequencies at position +1 in HEGs and LEGs are not significantly different ($p = 0.66$, Wilcoxon signed-rank test). In absolute terms the enrichment in LEGS is if anything higher. This is contra to the fail-safe prediction that ASCs should be most greatly enriched in HEGs.

Third, if the effect is owing to translation termination signals favouring +4T, then +4T enrichment might be expected to be most profound in TGA terminating genes and weakest in TAG terminating genes as RF2 crosslinking [43, 44] would be irrelevant for RF1-recruiting TAG. As TAA can use RF2 or RF1 it should be intermediate. To investigate this, we analysed the relative usage of thymine against adenine, cytosine, and guanine at the fourth site as a function of primary stop usage (Fig 4). Considering all genes this not only confirmed T enrichment compared to the next most frequent nucleotide, unique to TGA-terminating genes (T > A: $p < 2.2 \times 10^{-16}$, Wilcoxon signed-rank test) but, consistent with the RF2 crosslinking hypothesis, the +4T usage was in the order TGA>TAA>TAG. +4T frequency is significantly higher in TGA-terminating genes than TAG-terminating genes ($p < 2.2 \times 10^{-16}$, Wilcoxon signed-rank test) and TAA-terminating genes than TAG-terminating genes ($p = 7.5 \times 10^{-5}$, Wilcoxon signed-rank test).

The strength of +4T enrichment in TGA and weakness in TAG-terminating genes is underscored when we consider HEGs and LEGs separately. Thymine frequency at the fourth site significantly exceeded the next highest nucleotide regardless of the primary stop in HEGs, in the predicted order (T > A, $p = 2.9 \times 10^{-4}$ in TGA-terminating genes; $p = 0.013$ in TAA-terminating genes; $p = 0.045$ in TAG-terminating genes, Wilcoxon signed-rank tests). The signal in TAG-terminating genes in this instance does not withstand multi-test correction ($p > 0.05/3$). In LEGs, too, raw +4T frequency is found in the expected order TGA>TAA>TAG, with enriched frequencies of thymine evident only in TGA-terminating genes (T > A: $p = 1.2 \times 10^{-4}$, Wilcoxon signed-rank test).

## Analysis of T starting codons suggest a [TAA|TGA]T[T/C] motif

The above results suggest that any weak stop excess at codon position +1 is not owing to selection for stops *per se*. Is the enrichment for T-starting codons the same for all such codons, stops included, or might some classes be especially preferred, suggesting some further motif structures? To investigate this, we calculated an enrichment score for each T-starting codon (Fig 5). We notice an enrichment of TC and TT-starting codons at position +1, particularly in HEGs and TGA-terminating genes. Indeed, we propose that there may be a fifth nucleotide site preference for thymine or cytosine in +4T-containing genes as part of a wider motif beneficial for translation termination. Consistent with this, the enrichment of stop codons at position +1 is unremarkable compared to other T-starting codons. This is, too, consistent with our

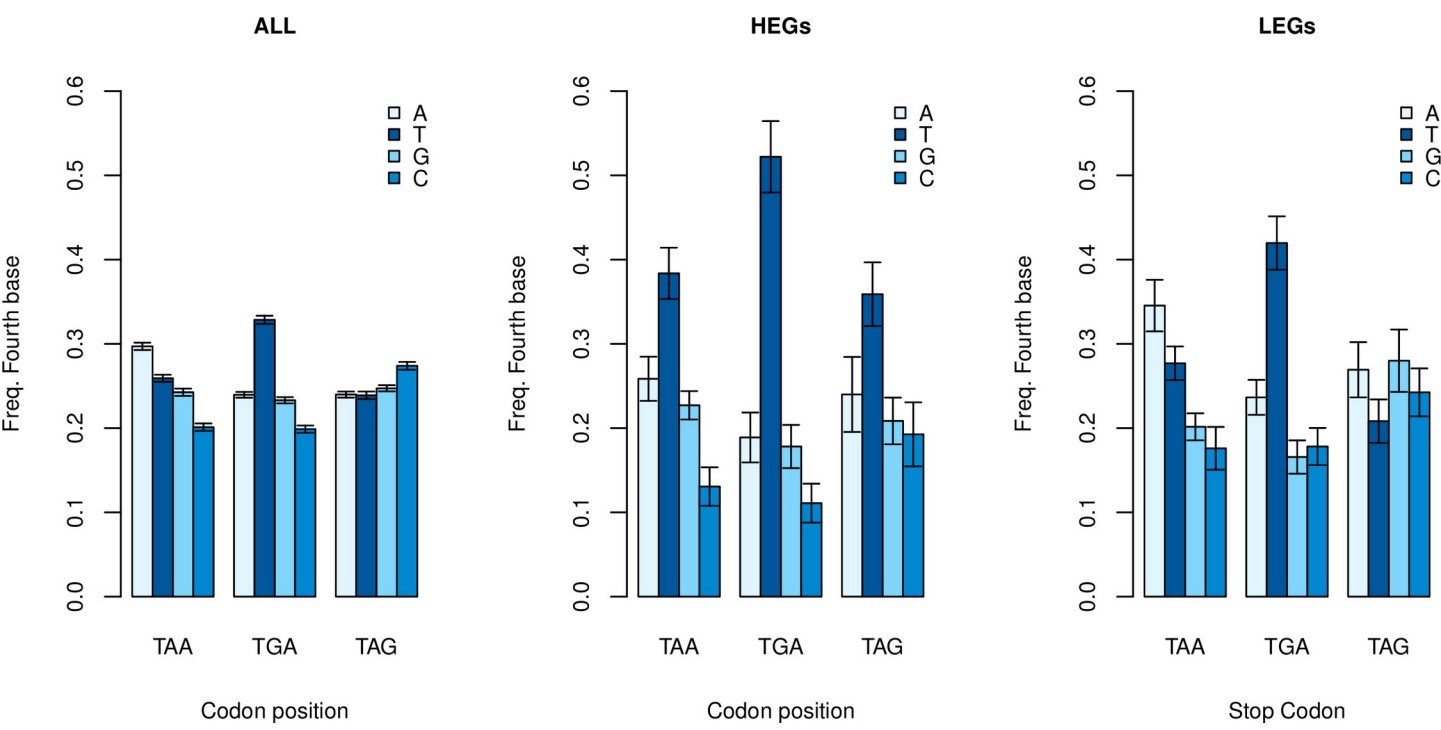

**Fig 4. Assessment of fourth base nucleotide frequencies as a function of primary stop usage.** Standard errors represent bootstrapped standard error. In all genes, not only is +4T enriched, compared to the next highest base, in TGA-terminating genes (T > A: p < 2.2 x 10$^{-16}$, Wilcoxon signed-rank test), but consistent with the RF2 crosslinking hypothesis, the +4T usage was in the order TGA>TAA>TAG.

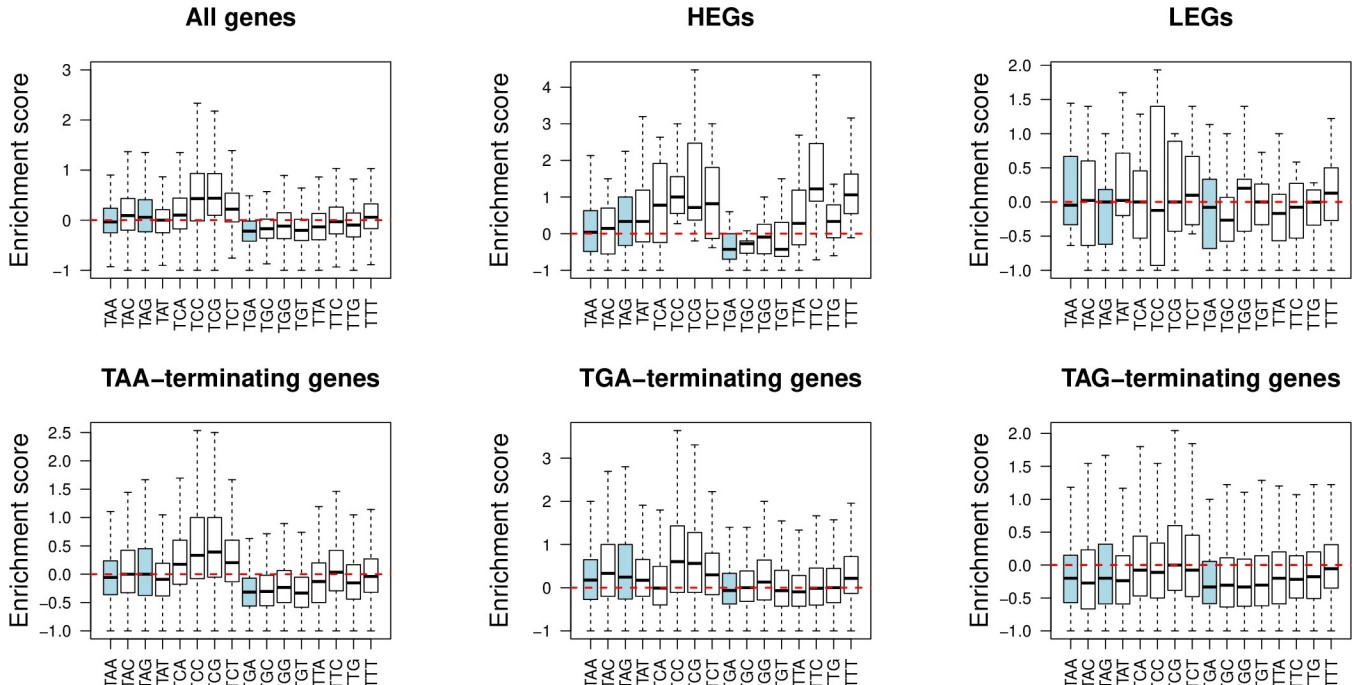

**Fig 5. Enrichment of T-starting codons at position +1.** Enrichment scores calculated for each T-starting codon at position +1 such that: Enrichment Score = [F1 / mean (F3 + F4 + F5 + F6)] -1, where F1 = frequency at position +1 etc. Stop codons, highlighted in blue, show no remarkable enrichment compared to other T-starting codons.

+4T-controlled simulation experiment (S5 Fig), which finds that increased ASC frequencies at position +1 are the direct consequence of +4T enrichment. Further analysis suggests that TT is preferred in HEGs regardless of the primary stop. This partially reflects an AT bias in our genome set and more generally the preference for TT in AT rich genomes and TC in GC rich ones (S6 Fig).

## No evidence that ASCs are enriched with downstream T, apart from TGA

As stops appear to prefer a +4T to enable stop codon recognition, we can ask whether this is also true of ASCs. We thus test the null that ASCs are as likely to have a downstream T as primary stops. For all genes, ASCs have significantly less chance to be immediately followed by a T than do primary stops ($p < 2.2 \times 10^{-16}$, Wilcoxon signed-rank test). The same is seen in HEGs ($p = 2.4 \times 10^{-6}$, Wilcoxon signed-rank test), though not LEGs ($p = 0.078$, Wilcoxon signed-rank test). The fail-safe hypothesis however does not necessarily predict selection termination functionality at ASCs to match that of primary stops. A more generous null is to ask whether ASCs have more T at the +4 site than do non-ASC codons in the 3' region. We actually find for all genes that ASCs have lower chance of having this ($p < 1.5 \times 10^{-12}$, Wilcoxon signed-rank test). The same is seen in both HEGs ($p = 7.0 \times 10^{-3}$, Wilcoxon signed-rank test) and LEGs ($p = 0.027$, Wilcoxon signed-rank test).

A more specific approach assesses each stop codon variant individually. As +4T enrichment appears to be peculiar to TGA-terminating genes, we expect TGAT to be more common as an ASC than TAAT, and even more so compared to TAGT. All three stop variants are significantly less likely to be followed by T when in-frame downstream than when located at the primary stop site (TAA $p < 2.2 \times 10^{-16}$; TGA $p < 2.2 \times 10^{-16}$; TAG $p < 4.6 \times 10^{-6}$, Wilcoxon signed-rank tests). Though whilst TAA ($p < 2.2 \times 10^{-16}$, Wilcoxon signed-rank test) and TAG ($p < 2.5 \times 10^{-12}$, Wilcoxon signed-rank test) are less likely to possess a 3' neighbouring T than non-ASC codons, TGA is significantly more likely to ($p < 2.2 \times 10^{-16}$, Wilcoxon signed-rank test). Hence there is exceptionalism of TGA which falls in line with the expectations of the fail-safe hypothesis. Indeed, ASC +4T frequencies are found in the expected pattern TGA > TAA > TAG (TGA > TAA $p < 2.2 \times 10^{-16}$; TGA > TAG $p < 2.2 \times 10^{-16}$; TAA > TAG $6.0 \times 10^{-5}$, Wilcoxon signed-rank tests). We do, however, find a contradictory result in that TGAT is no more common in HEGs than LEGs ($p = 0.56$, Wilcoxon signed-rank test), though this is affected by low genome sample sizes.

One might suggest that the enrichment of T following TGA in 3' positions compared to other non-stop codons could be attributed to dinucleotide preference. We control for this by comparing 3' TGA to non-stop codons with third nucleotide A, finding again that TGAT to be significantly more common ($p = 2.9 \times 10^{-5}$, Wilcoxon signed-rank test).

## No evidence that ASC presence predicts reduced downstream ASC frequency

The above analyses provide little support for the fail-safe hypothesis as any weak site +1 trends appear better explained by +4T motif presence. The observation of ASC enrichment at codon +2 in TGA terminating HEGs (sensitive to Bonferroni correction) and the enrichment of 3' TGAT are the only results that doesn't obviously fit with this otherwise profound rejection of the hypothesis. Given this, and the difficulties allowing for complex GC pressure and motif issues, we consider alternative tests.

In theory, if ASCs function in the termination of translation, it is unlikely that an ASC will be followed by another. The combined action of the primary stop and the ASC should terminate translation such that net readthrough rates are negligible and there is no selection for a

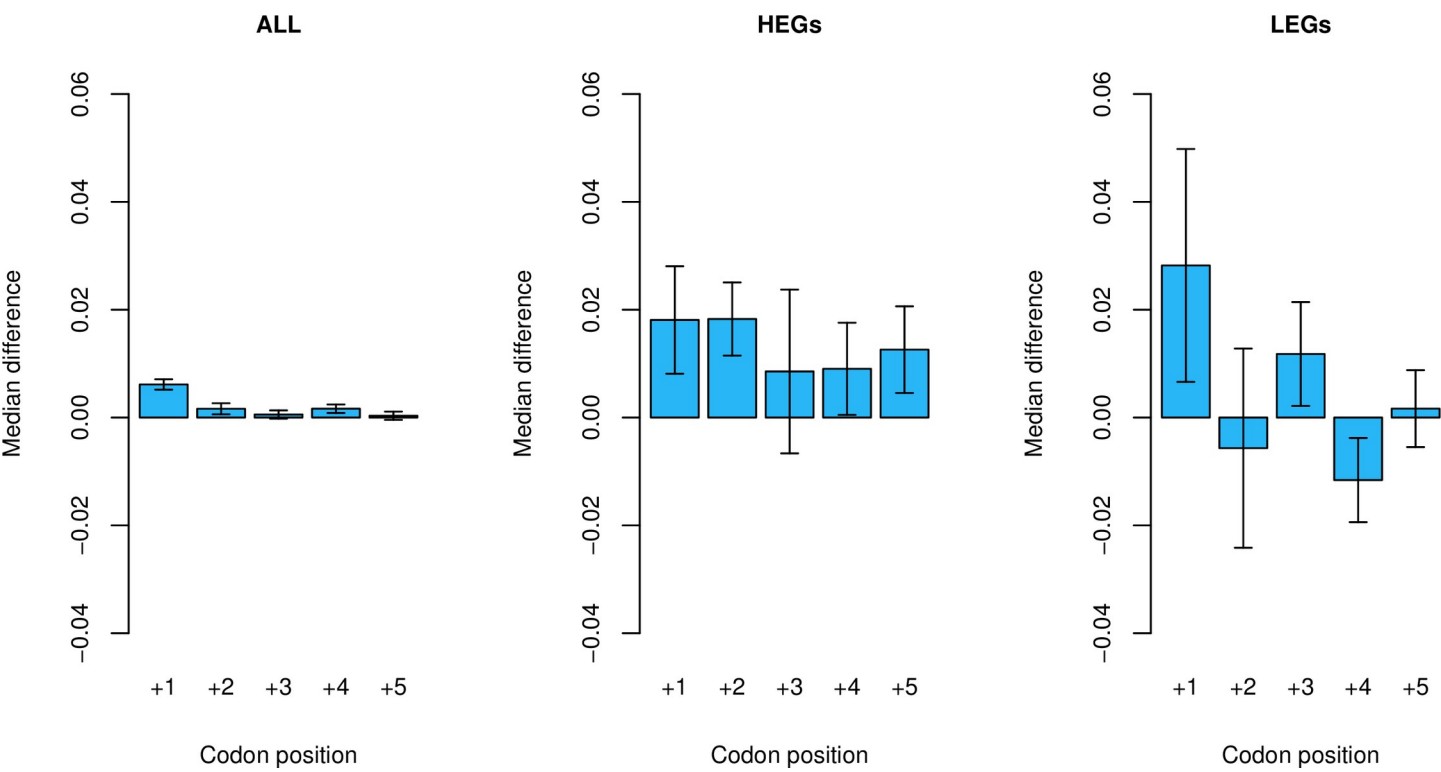

**Fig 6. ASC-containing genes (at position +N, where N is a downstream codon position from +1 to +5) compared against ASC-absent genes for the presence of another ASC at the next codon position.** Bars represent the median difference between ASC-absent and ASC-containing genes at each focal position. Error bars represent bootstrapped standard error. In the 'all genes' group, where an ASC is present at position +1 there is a significantly reduced chance of having another ASC at position +2 (Wilcoxon signed-rank test: p = 3.6 x 10$^{-3}$).

third stop. We thus test the null hypothesis that ASC-containing genes, where the stop codon lies before (and including) codon +N, have an equal chance of possessing a further ASC downstream. We compare downstream ASC frequencies of ASC-containing and ASC-absent genes and see no evidence that possession of a stop predicts low rates of downstream stops (p = 0.83 where the focal codon is position +1, p = 0.76 for position +2, p = 0.77 for position +3, p = 0.78 for position +4 and p = 0.92 for position +5, one-tailed Wilcoxon signed-rank tests). This provides no support for the fail-safe hypothesis.

We can ask a more detailed question, namely whether having a stop at position N predicts the absence of a stop at the next codon position (+N+1, rather than generically downstream). In this case 'N' refers to each position from +1 to +5 (position +6 could not be tested in this instance as this would require analysis of ASCs at position +7, which is not considered). Where we consider all genes, ASC-absent genes demonstrate no significant excess of ASCs at position +N+1 over ASC-containing genes at all positions (p > 0.05), except where the focal codon position was position +1 (p = 3.6 x 10$^{-3}$, Wilcoxon signed-rank test; Fig 6). Were this owing to selection, we expect to find a stronger signal in HEGs than in LEGS. However, there is no significant difference between HEGs and LEGs at any position (p > 0.05 for all positions +1 to +5, Wilcoxon signed-rank tests). A significant signal can only be found in HEGs at position +2 (p = 0.027, Wilcoxon signed-rank test), although this result does not survive multi-test correction (p > 0.05/5). We do however notice that the magnitude of the effect is actually greater in LEGs, which is contra to the fail-safe hypothesis.

We conclude that these tests provide no robust evidence that the presence of a stop codon predicts the presence/absence of further stops and if any such effects exist they are specific to

the domain in the immediate vicinity of the primary stop, suggesting that hidden motifs might be a viable alternative explanation.

## TGA is under-used in TT4 mollicute UTRs compared to GC matched genomes that use TGA

The mollicute bacteria provide for a "natural" experiment as some genomes employ TT4 in which only TAA and TAG are used for chain termination. Hence, as TGA functions as a stop codon in TT11 genomes, it is expected under the fail-safe hypothesis that TGA frequency 3' of the primary stop in TT4 genomes should be consistently lower than that in TT11 genomes.

We tested this hypothesis by fitting a LOESS model (span = 2/3) for positions +1 to +6 usage of TGA against genomic GC3 in TT11 genomes of the full genome set. These models allowed the prediction of TGA frequencies of TT4 mollicute genomes at each position given their genomic GC3 content. TGA frequency was significantly reduced in TT4 genomes compared to predicted by the LOESS model at positions +3 and +5 ($p = 1.5 \times 10^{-1}$ for position +1; $p = 9.8 \times 10^{-2}$ for position +2; $p = 9.7 \times 10^{-4}$ for position +3; $p = 7.9 \times 10^{-1}$ for position +4; $p = 3.4 \times 10^{-4}$ for position +5; $p = 7.7 \times 10^{-2}$ for position +6, Wilcoxon signed-rank tests). For comparison, TT11 mollicute genomes do not significantly under-use TGA at any position ($p > 0.05$, Wilcoxon signed-rank tests). In TT4 genomes, lack of underrepresentation at position +1 possibly accords with the utility of +4T and similar motifs adjacent to the primary stop. The poverty of TGA at positions +3 and +5 survives multi-test correction and is consistent with the possibility that TGA maintains a function in TT11 genomes beyond its role in TT4 genomes. Why TGA is not underused at positions +2, +4 and +6 is unexplained. We do, however, find that when considering the whole UTR (positions +1 to +6) TGA is used significantly less often in TT4 genomes than predicted ($p = 3.8 \times 10^{-6}$, Wilcoxon signed-rank test). We acknowledge the limitations of LOESS modelling, which include those relating to the arbitrary nature of kernel/span function, and therefore validate this result with a different test design (S7 Fig). Given the above we also asked whether TAA, TGA, and TAG codon switches occur at different rates in TT4 genomes. We find no significant differences (S2 Table) but strongly caution that the results are limited by drastically reduced gene sample size.

The above results are consistent with the hypothesis that TGA is underused in 3' domains when it isn't employed as a stop codon, compared with its usage in genomes of similar GC content when it can function as a stop. However, if TGA is underrepresented in TT4 decoded genomes due to its selection for error-proofing in TT11 decoded species, we expect the magnitude of this under-enrichment to consistently surpass all other codons. We thus investigated all 64 codons using the same LOESS methodology and ranked them by their one-tailed Wilcoxon signed-rank test p-value (S3 Table). We find TGA to be just the 25th most under-enriched codon at position +1, 20th at position +2, 4th at position +3, 49th at position +4, 2nd at position +5, and 16th at position +6. Instead, we find codons CCG (1st at positions +1, +4, +6), GTG (2nd at position +1, 3rd at positions +4 and +6), and TAT (1st at position +2, 2nd at position +3, 4th at position +1) among the more commonly underrepresented codons at specific positions. Assertions that there is something special about TGA, specifically relating to translational termination, therefore remains speculative.

## Between-genome primary stop codon usage is reflected in downstream positions

The disconnect between TAG and TGA usage as a primary stop has been attributed to co-evolution between RF1:RF2 ratios and GC content [14]. If true, this renders stop usage tightly

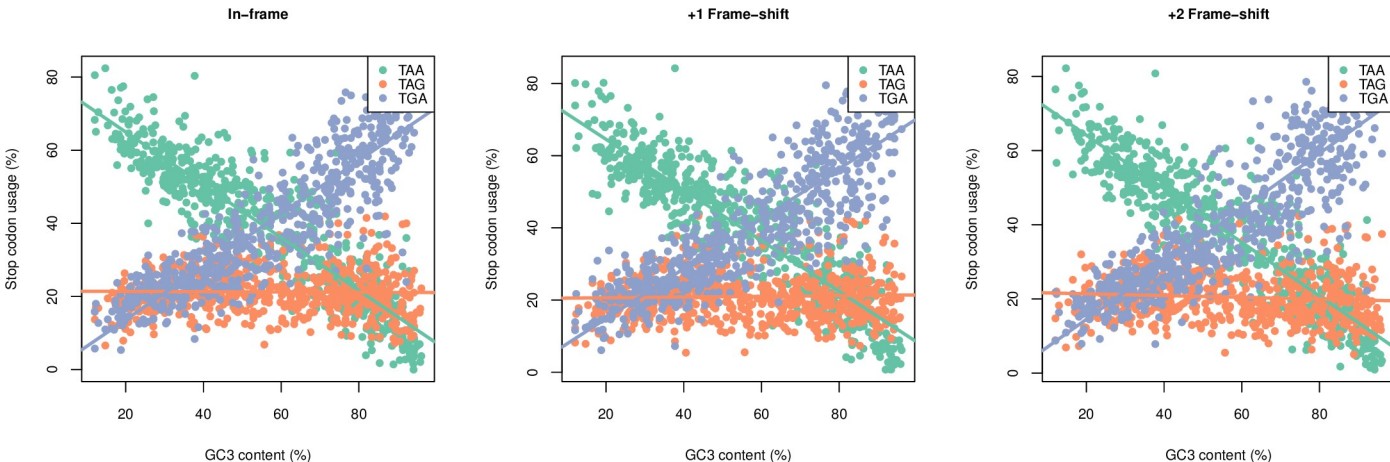

**Fig 7. Relative usage of each stop codon in 3' UTRs plotted against GC3 content for TT11 bacterial species.** Surprisingly, we find that trends in the decoupled TGA and TAG to be consistent across all three reading frames. Spearman's rank correlation information can be found in S5 Table.

coupled to the mechanistic basis of translational termination. Are then these trends in TAA, TGA and TAG usage also seen downstream?

First, we analysed the relative usage of TAA, TGA and TAG at the primary site so as to repeat the findings of Korkmaz and colleagues (2014) with our genome set (S8 Fig). As expected we find that TAA-usage is negatively correlated with genomic GC3 ($\rho$ = -0.92, $p < 2.2 \times 10^{-16}$, Spearman's rank correlation), TGA-usage is positively correlated with genomic GC3 ($\rho$ = 0.88, $p < 2.2 \times 10^{-16}$, Spearman's rank correlation), and TAG-usage shows no significant correlation and remains at low levels regardless of genomic GC3 ($\rho$ = -0.017, $p$ = 0.663, Spearman's rank correlation). We then returned our focus to downstream. Surprisingly, we find that trends in TGA and TAG usage remains clearly decoupled despite their equal GC content. Indeed, trends in stop codon usage are remarkably similar between positions +0 to +6 (S4 Table).

That stop codon usage at the primary stop consistent in 3' positions implies either a) that the release factor hypothesis [22] regarding the decoupled usage of TGA and TAG usage is wrong or b) ASCs are, despite all the other negative data, under selection as fail-safe codons. We can investigate this by considering all three reading frames: should the relative codon usage of ASCs remain consistent in +1 and +2 frame-shift environments we can be relatively confident that usage is not controlled by selection relating to translational readthrough or termination. This is exactly what we find (Fig 7), and this is consistent with the bulk of the evidence described in our study. Thus, we suggest that the RF1:RF2 ratio is not the correct explanation for the differential stop usage as a function of GC and we are instead missing some important information regarding TGA and TAG usage.

## Lack of ASC enrichment in bacterial genomes reveals a discrepancy with single-celled eukaryotes

The above bacterial evidence against ASC enrichment is in contrast to that seen in yeast and ciliates [39, 40]. Do prokaryotes and eukaryotes truly differ in their propensity to use ASCs to control translational read-through rates? Alternatively, might there be a reporting bias in which only significant effects surface in the published literature, thereby giving a skewed view of the commonality of fail-safe stops? Additionally, there are several ways to evaluate the fail-safe hypothesis and it could be that our methods would fail to report effects in the eukaryotic

species within which ASC enrichment has been observed. For example, while we employ a dinucleotide control, Adachi and Cavalcanti in the prior ciliate analysis [40] employ a method that considers the rate of occurrence of the first 3’ stop as a function of downstream position given an underlying rate at which stops are observed in 3’ UTR.

To ask whether our method would recover enrichment where previously claimed, we consider ASC enrichment in *T. thermophila*, *P. tetraurelia* and *S. cerevisiae* via the calculation of Z-scores, i.e. using the same method described earlier (Fig 8). Significant enrichment ($Z > 1.64$, for one tailed test of enrichment) is detectable using whole 3’ UTR frequencies in the two ciliates but not in yeast. The latter negative result is not surprising as, unlike in ciliates, yeast enrichment is only detectable at position +3 and predominantly only when the primary stop is TAA [39]. Indeed, we find that position +3 is unusual in being enriched ($Z>0$) in ASCs in all genes and in TAA-terminating yeast genes (Fig 8), although in neither is the effect significant ($Z = 0.93$ for all genes, $Z = 0.70$ for TAA-terminating genes).

These results suggest that our method can capture some but not all of the prior claims. Nevertheless, we extend the Z-score analysis to a set of 68 single-celled eukaryotes to investigate whether the spread of Z-scores matches that of the bacteria. We propose that single-celled eukaryotes are the fairest comparators to eubacteria as they are likely to both have large effective population sizes and, being single celled, would suffer the immediate consequences of any fitness costs of read-through. Multi-cellular organisms, by contrast, might be able to buffer fitness loss in one cell, for example by apoptosis and cell replacement. A genome is considered ‘enriched’ if it contains significant ASC enrichment at one or more positions ($Z > 2.33$, Bonferroni corrected one tailed). Interestingly, we find 20/68 of our eukaryotic genomes to be enriched, compared to 0/644 of our bacteria, these proportions being significantly different ($p < 0.0001$, $\chi2 = 184.3$, Chi$^2$ test).

An alternative metric is to consider the number of genomes showing enrichment, defined by chi-squared, above dinucleotide controlled null frequencies at each position. For this we

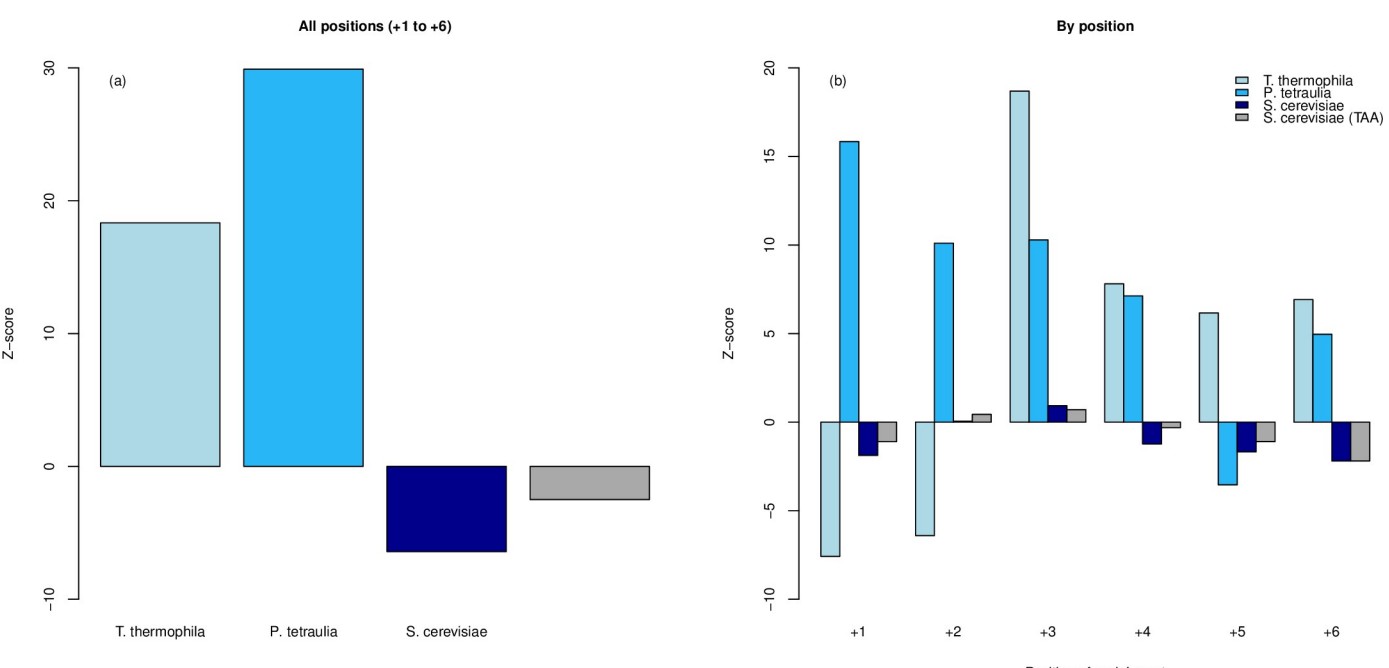

**Fig 8. Z-score analysis of previously analysed eukaryotic genomes.** Z-scores representing deviation from dinucleotide controlled null simulations over the whole 3’ UTR sequence (a) and then each position individually (b) for three eukaryotic genomes.

employ a Chi$^2$ p-value < 0.05/$n$, where $n$ is the number of positions tested, and apply this to our TT11 bacterial set of genomes and our set of 68 single-celled eukaryotes. Having defined positional ASC enrichment as p < 0.01 (0.05/5) as we analyse five positions (+2 to +6), the probability of a genome not possessing significant ASC enrichment at one or more positions is $0.99^5$ (approximately 0.951). There is hence a $1-0.99^5$ (approximately 0.049) probability that a genome will contain significant enrichment at one or more positions. Hence, our null is that 4.9% of our genomes are expected to show ASC enrichment by chance alone. In our eukaryotic set, we find over-representation of genomes containing significant ASC enrichment compared to this null prediction (21/68, p = 6.12 x 10$^{-12}$, one tailed-binomial test with p = 0.049, expected = 3). Such a result supports evidence for ASC enrichment in eukaryotic systems [39, 40], however we note that whilst ASC enrichment is commonplace, it is not universal nor consistent in its position. By contrast in bacteria, using this same method, we find that significantly fewer bacterial genomes show enrichment than expected by null (21/644, p = 0.028, one-tailed binomial test with p = 0.049, expected = 32, Fig 9), consistent with a broad claim that eubacteria seem to avoid ASCs. Moreover, the observed proportions of 21/644 in bacteria and 21/68 in eukaryotes are significantly different (p < 0.0001, χ2 = 79.6, Chi$^2$ test), corroborating the results of our Z-score analysis.

We also repeat the Chi$^2$ comparison using an alternative null model as proposed by Adachi and Cavalcanti [40]. This too confirms the same results (Fig 10), namely avoidance of ASCs in bacteria, enrichment in single-celled eukaryotes. Indeed, this mode of analysis reports enrichment at one or more positions in 32 of 68 eukaryote genomes and only 7 of 644 bacterial genomes, these proportions being different (χ2 = 242.3, p < 0.0001, Chi$^2$ test).

The conclusions that there is indeed a discrepancy between bacterial and eukaryotic propensity to select for ASCs is hence both real and largely resilient to methodological nuance. With respect to the eukaryotes, we corroborate significant ASC enrichment (using at least one methodology) in the previously analysed yeast [39] (*S. cerevisiae*, plus *C. albicans*) and ciliates [40] (*P. tetraurelia*, *T. thermophila*). We note that the two ciliate species analysed in the prior study [40] possess a re-assigned translation table (TGA is the only stop). We not only recover ASC enrichment in these re-assigned ciliates (plus *S. lemnae*), but a translation table 11 (TGA, TAA and TAG are all stops) ciliate as well (*S. coeruleus*). Of our methodologies, the two dinucleotide-controlled analyses (Z-score: 20 enriched genomes, S7 Table; Chi$^2$ analysis: 21 enriched genomes, S8 Table) appear to be the most stringent in detecting eukaryotic ASC enrichment. Identification of enrichment using the Adachi and Cavalcanti null model [40] is more generous (32 enriched genomes, S9 Table). We do, however, note that ASC enrichment at one or more positions is recovered by all three methods in 15 eukaryotic genomes, indicating reasonable overlap between the tests.

## Discussion

Our results suggest that, unlike in yeast, ciliates and some other protists, the error-proofing role of ASCs in bacteria is minimal at best. We began by testing the most obvious prediction of the fail-safe hypothesis, that stop codons should be enriched downstream of the primary stop codon. Having found no evidence for this at a genome-wide level, we considered the conservation of ASCs and found evidence that stops are less preserved than expected, this too being consistent with apparent avoidance. Additionally, we compared highly expressed and lowly expressed genes, seeing no differences. Comparing TGA-terminating HEGs and TAA-terminating LEGs we found TGA-terminating HEGs do not contain significantly higher ASC frequencies, except at position +1. The effect seen at +1 is not the result of selection for stops, but rather a knock-on consequence of selection for T-starting codons at the first codon

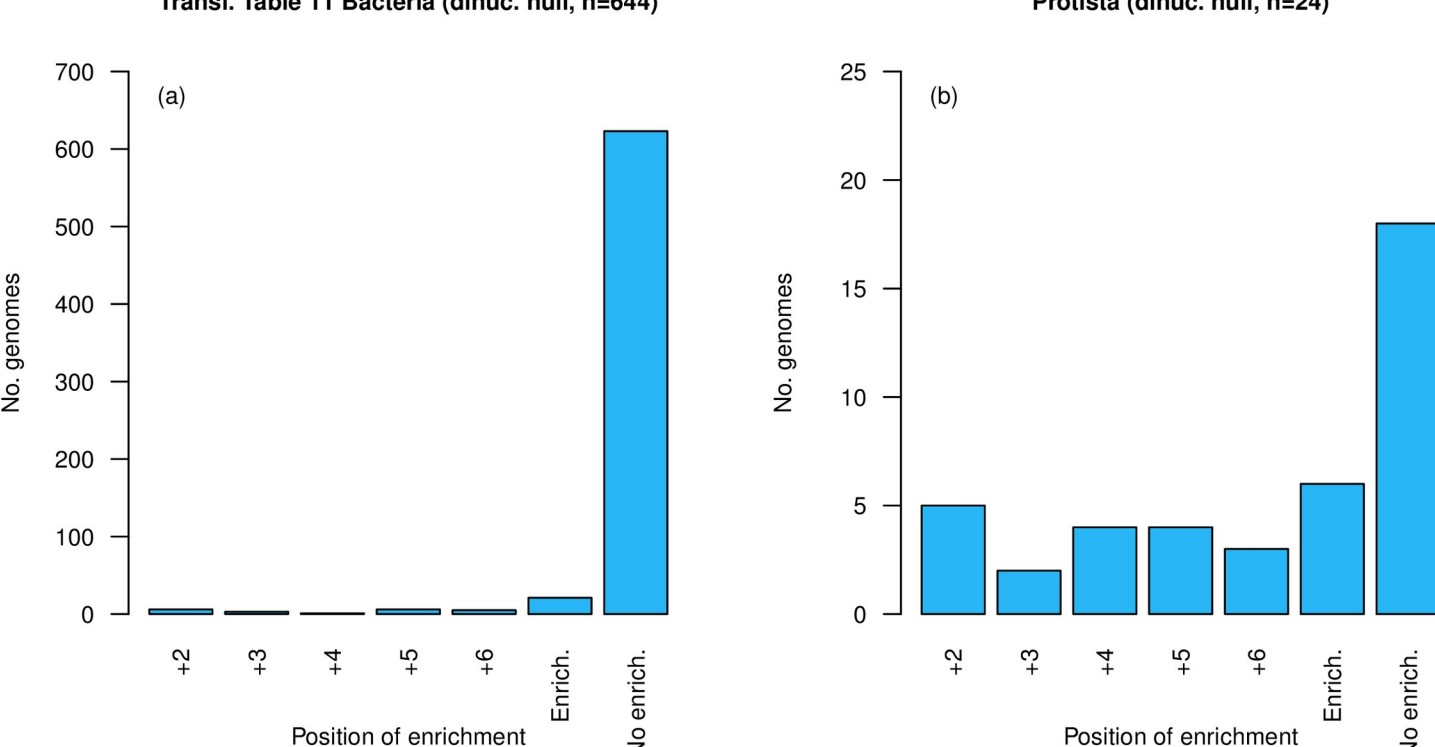

**Fig 9. Number of genomes showing enrichment over dinucleotide-controlled null at each position, excluding position +1, in two genome sets (a) translation table 11 bacteria (n = 644) and (b) single-celled eukaryotes (n = 68).** Genomes showing enrichment are underrepresented in the bacterial set (21/644, p = 0.028, one-tailed binomial test, expected = 32) and overrepresented in the eukaryotic set (21/68, p = 6.12 x 10$^{-12}$, one tailed-binomial test, expected = 3). 'Enrich' is the total number of genomes with enrichment at one or more positions. 'No enrich.' is the total number of genomes with no enrichment at any position.

downstream, the trend being seen for non-stop T-starting codons too. Indeed, in the context of other T-starting codons stop codon usage is not simply unremarkable, the trend seems to be enrichment for non-stops, TT and TC being preferred residues. While it is suggestive that the leakiest codon (TGA) is the one associated with ASC enrichment at site +1, this trend is better explained by reference to the notion that RF2 cross-links with the adjacent +4T and TGA uses only RF2. Perhaps an informative test would compare species with defective/absent RF2 to those without, however we find no such genomes in our genome set. These results suggest bacteria and eukaryotes are different in the usage of fail-safe stops. Using several alternative methodologies to compare ASC enrichment in bacteria to protists, we validate that ASC enrichment is found in single celled eukaryotes more often than in bacteria. Our findings therefore highlight a discrepancy in the way that bacterial and eukaryotic genomes evolve in response to translational read-through. With respect to bacterial transgenes, our results thus do not support any major adjustments to their design or experimental protocols, beyond using TAA or TAAT[T/C] for termination.

A few results were consistent with the fail-safe hypothesis but not overwhelmingly so. While having a stop codon at any given position doesn't predict a dearth of downstream stops, if there is a stop at position +1 there is less likely to be one at position +2. However, the magnitude of this effect is greater in LEGs than HEGs questioning the overall relevance of this to the fail-safe hypothesis. Given too that the effects are seen exclusively in proximity to the primary stop, selection on unrecognised motifs is a viable and probably better alternative explanation. That TT4 mollicutes contain fewer TGAs in their 3' domain than expected is also enigmatic.

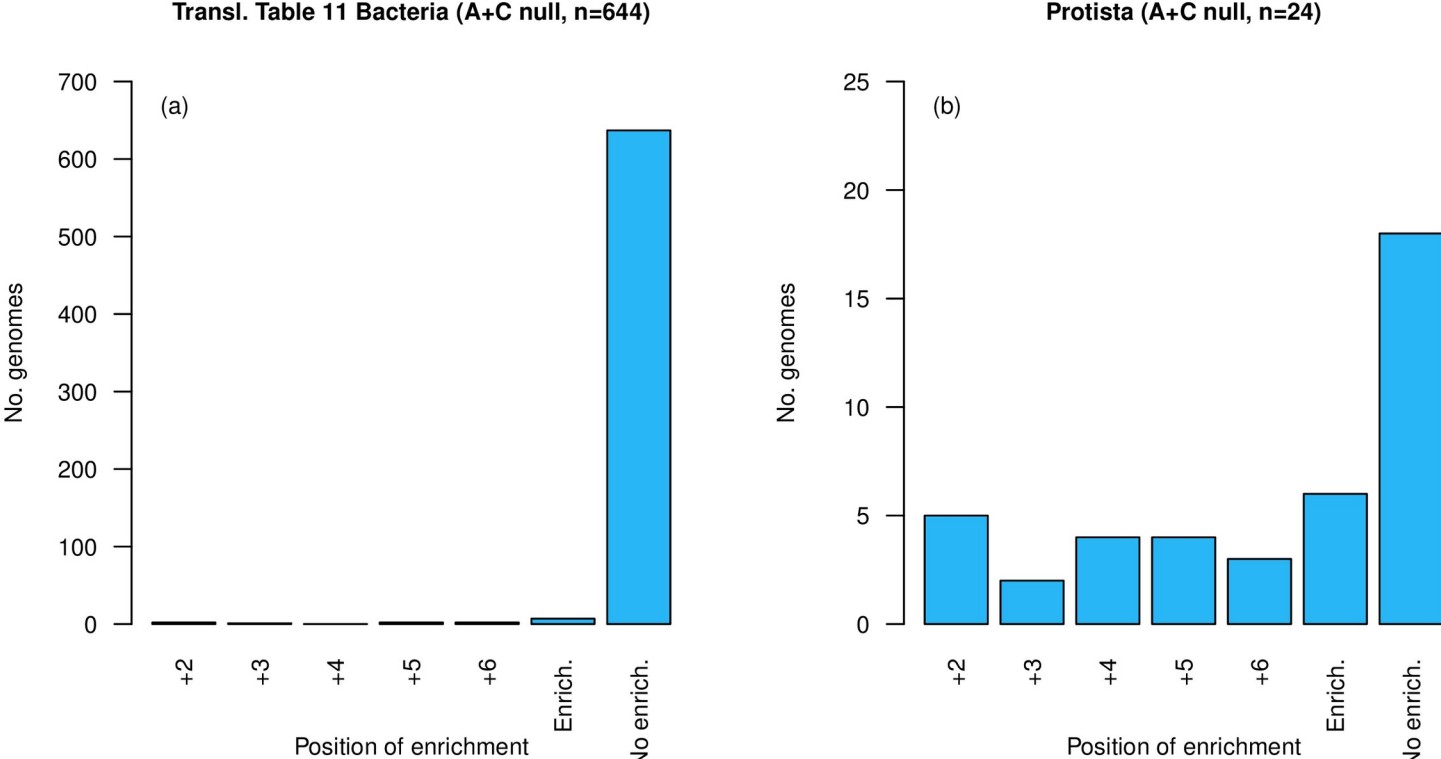

**Fig 10. Number of genomes showing enrichment over A+C null (see methods) at each position, excluding position +1, in two genome sets (a) translation table 11 bacteria (n = 644) and (b) single-celled eukaryotes (n = 68).** Genomes showing enrichment are underrepresented in the bacterial set (7/644, p = 9.5 x 10$^{-8}$, one-tailed binomial test with p = 0.049, expected = 32) and overrepresented in the protists set (32/68, p = < 2.2 x 10$^{-16}$, one tailed-binomial test with p = 0.049, expected = 3). 'Enrich' is the total number of genomes with enrichment at one or more positions. 'No enrich.' is the total number of genomes with no enrichment at any position.

That TT4 mollicutes contain less 3' UTR TGA than TT11 genomes (after control for GC content) is consistent with selection impacting TGA levels in 3' domains of TT11-decoded species. However, in TT11 genomes we see no evidence for ASCs beyond null levels and indeed, *prima facie* they seem to be avoided more often than enriched compared to GC controlled nulls. Furthermore, that some sense codons are even more consistently under-used at 3' UTR sites, for reasons that are unknown, suggests that there is a gap in our knowledge of the biology of these 3' ends.

A third possibly consistent result is that ASC usage of the three stops as a function of GC content matches that of the primary stop. The patterns for the primary stop were speculated to reflect co-evolution between GC content and RF1:RF2 ratios [14] but this remains to be verified. That we see the same broad trends at all downstream positions, despite all the other evidence against these ASCs being functional stops, we suggest more profoundly questions the RF1:RF2 ratio model than it supports ASC functionality. In accord with the under usage of TGA and other codons in TT4 genomes, perhaps more complicated dinucleotide or trinucleotide preferences should be considered.

This leaves one outstanding observation, namely that 3' TGA tend to be followed by T more than expected, even given the rate of T-starting codons with an A immediately prior. How can we explain this? We suggest a hypothesis that might explain the curious observations against a backdrop of a large body of negative evidence. First, we wish to discount the possibility that the lack of evidence for selection on ASCs relates to read-through not being a strong enough selective force. Experimental estimates in *E. coli* and *S. typhimurium* suggest that the

read-through rates are really very high. A read-through event at a TAA-terminating site can occur at frequencies between >1 x $10^{-5}$–9 x $10^{-4}$ [29], and at a TAG between 1.1 x $10^{-4}$–7 x $10^{-3}$ [28, 29, 32, 33]. If ASCs do meaningfully function in chain termination, one would have expected to find a signal in TGA-terminating genes, where readthrough may occur at rates of 1 in 1000 translation events up to 1 in 100 [15, 30, 31]. Thus, numbers suggest a potentially high rate of readthrough. Second, it is most likely because of this that stop codons are themselves subjected to selection for efficient termination. This is probably why TGA-terminating genes are rarely highly expressed–where such selection is expected to be strongest and TAA is over-represented in the set of highly expressed genes even in GC rich genomes [13]. Consistent with this, Belinky and colleagues (2018) found that stop codon switches occur significantly more frequently than the equivalent substitutions in non-coding DNA. Given this we assume that selection against read-through is a significant force.

We can then question whether, if read-through is the problem, ASCs are the expected solution in bacteria. Evidence from stop codon usage, especially in highly expressed genes suggests that there is selection for TAA enrichment as the stop. We could presume that in many cases this means simply a non-TAA stop mutates to be TAA and is selectively favoured, especially if the gene is highly expressed. However, there are other possibilities. For example, imagine that we have a highly expressed gene using TGA and so possesses high read-through rates. Imagine too that upstream are sense codons which could mutate in one step to TAA or indeed any stop. This would introduce a premature stop (assuming the context is otherwise fine) with, importantly, a guaranteed fail-safe stop downstream i.e. the original primary stop. There would be a benefit from lower net read-through rates (we presume nearly all genes will terminate at or before the second, original, stop) and a benefit from reduced translation costs when the new earlier stop functions. Moreover, the sequence now immediately 3' of the new stop will, if read-through happens, be sense codons of a recently functional protein, so there should be no toxicity of this additional sequence. All of these benefits suggest this is a viable path for evolution, the major cost owing to reduced gene length affecting protein function. However, tolerating such a cost appears to be possible, with stop codon shifts in 5' directions now thought to have an under-appreciated influence on gene shortening [58]. If the net benefits of reduction in read-through is greater than this cost then the system will have evolved towards reduced net readthrough.

Could this also explain why we detect no enrichment of stop codons in the 3' domain as, until the first ASC, selection would have recently been on this to perform as coding sequence? Might this explain the apparent general rarity of downstream T following an ASC? A stop lacking the +4T would be especially leaky and so especially favour rescue by creation of an earlier stop. The one exception could be TGAT. If this, like TGA, remains relatively leaky (unlike TAAT) then selection could still favour 3' stop creation. Might this also go some way to explaining the mollicutes result? If TGA wasn't a stop there is no reason it would by necessity feature in the 3' domain as the abandoned stop and so might appear at low frequency in the mollicutes.

An alternative trajectory to rescue a leaky TGA would be for TGA to mutate but to a sense codon. This could be favoured if the run on then meets a less leaky stop codon shortly downstream. The shortening process we suggest would be more common than the lengthening for several reasons. First, especially in highly expressed genes, addition of amino acids is likely to be costly, whereas loss would come with an energetic saving. Second, in the shortening process there are multiple potential sites that could mutate to a new upstream stop, while in the latter the mutation is required at the stop codon. Third in the gene shortening mode, at the time of mutation, at least one downstream site will be an ASC (the old primary stop), thus the system comes with guaranteed ASC protection. By contrast, gene extension could replace a leaky stop

with, at best a less leaky stop, but no guaranteed fail-safe ASC. Fourth, there is no guarantee on extension that the extension isn't toxic, while for read-through after shortening this would not be an issue. Thus, we suggest there may be a process to shorten highly expressed genes to enable evolution of protection from read-through that might be particular to prokaryotes. The difficulty with this model seems to be that the rate at which this would need to occur might have to be rather high. Whether this predicts any pattern is unclear as genes cannot continue to shorten indefinitely.

Might a propensity to gene shortening as a mechanism to cope with read-through also explain why ASC enrichment isn't seen in bacteria but is in eukaryotes? In eukaryotes the mutation creating this new upstream stop could be trapped by eukaryote-specific nonsense mediated decay (NMD) making gene shortening a non-viable solution. Perhaps for eukaryotes ASCs are the only viable solution (although how NMD knows a 3' stop isn't the true stop and the real primary stop not a premature stop is unknown). The model is consistent with HEGs generally being shorter (S6 Table) but this is not a discriminating prediction as a simple trans-lational cost argument would predict the same. Arguing against such a model however is the finding that stops in the vicinity of the true stop might not trigger NMD, the stops having to be 3' of the last intron, at least in some species [59–61].

An alternative possibility to explain the eukaryote-prokaryote divide concerns the possibil-ity that in some eukaryotes read-through rates can be greatly increased. Notably, the yeast prion [PSI+] state has been linked to extensive read-through via the misfolding of release fac-tor Sup35p [6, 62, 63]. It is tempting to speculate that this provides a possible mechanism for increased selection of ASCs in yeast not seen in bacteria. Though the [PSI+] system in yeast is possibly best studied, it now appears that prion-like systems are present throughout the tree of life [64, 65], including bacteria [64]. Not all prion-like states affect translation termination, however. The identification of species susceptible to prion-induced increased translational read-through rate could provide a means to test the fail-safe hypothesis in the future. Such a model predicts co-incidence between genomes with ASC selection and prion-like systems affecting translation. Indeed, we are unaware of any bacterial prion system disrupting transla-tional termination which would be consistent with the absence of ASC selection. The closest resemblance that we are aware of is with a system in *Clostridium botulinum* affecting a domain of transcription (not translation) termination factor Rho (*Cb*-Rho) [66].

Above we have presumed that read-through rates are the same in all genomes, with the pos-sible exception of prion mediated read-through. In this context we note a further striking peculiarity, that ASC rates (Z-score deviation from dinucleotide controlled null) are especially low in GC-rich organisms. GC-rich organisms are typically thought to be those with stronger selection as the underlying mutational bias is towards AT [67, 68]. Assuming this reflects higher effective population sizes in GC-rich organisms, the lower Z-scores in GC-rich organ-isms is enigmatic—if anything one might expect selection to favour more ASCs if selection is strong. It is also enigmatic as in GC-rich genomes the span to the next random stop in the 3' domain is likely to be longer as stops are AT-rich, hence GC-rich genomes should also be under selection to conserve ASCs. However, this assumes all else is equal. If AT-rich bacteria are subject to higher read-through rates, the GC-trend might make some sense. Such a model would fit in the broader context of the possibility of stronger selection against error creation when populations are large and selection efficient [69]. Comparably, GC-rich organisms have a broader spectrum of tRNAs thought to reduce ribosomal frameshifting rates [70]. Might this also reduce read-through rates? An alternative possibility is that in GC-rich genomes, random ASCs are less likely to function as stops. If for example AT-richness in the vicinity of a stop is needed to enable stop functioning, then a random ASC in a GC-rich genome is, for example, unlikely to have a +4T and might thus be ineffective. Indeed, experimentally tandem stops

appear not to have the expected level of read-through suggesting particular context requirements [41, 42]. We suggest that experimental determination of read-through rates in organisms with different tRNA profiles would be informative.

## Methods

### General methods

All analyses were performed using bespoke Python 3.6 scripts. Statistical analyses and data visualisations were performed using R 3.3.3. Scripts can be found at https://github.com/ath32/ASCs. Whilst it is acknowledged that stop codons function at the mRNA level, in this analysis we have analysed chromosomal DNA sequences and henceforth refer to the three stops as TAA, TGA and TAG and to +4U enrichment as +4T. Please note that in all other contexts +1, +2 etc refer to the position of downstream codons, not nucleotides, with +1 being the codon immediately after the primary stop.

### Genome downloads and filtering

Whole-genome sequences for 3,727 bacterial genomes were downloaded from the European Molecular Biology Laboratory (EMBL) database (http://www.ebi.ac.uk/genomes/bacteria.html, last accessed 1st August 2018). For the majority of the analyses, genomes were filtered to include only one genome per genus, so as to prevent over-sampling from the very well surveyed groups and hence to reduce any bias attributable to phylogenetic nonindependence. So as to exclude plasmids, incomplete genomes or very small genomes we retained only those genomes larger than 500,000 base pairs. This generated a sample of 650 genomes, 644 that employ translation table (TT) 11 and 6 using TT4, in which TGA no longer functions as a stop. The exception to this filtering was the specific analysis of mollicute and TT4 genomes, which were filtered directly from the raw sample of 3,727 genomes (106 and 94 genomes respectively). Of these genomes, only those with > 100 genes were considered for analysis.

For every gene in each genome, a sequence inclusive of the primary stop followed by 27 nucleotides of the 3' UTR was extracted by applying coding sequence coordinates to the total genomic sequence attainable in the EMBL files. Only genes with 3' intergenic space of >30 base pairs were considered for analysis, thus ensuring a sample of genes with sufficient 3' UTR length. Resultant sequences were filtered to retain only those 3' sequences made up exclusively of A, T, G and C, those from genes with one stop after the initiating codon, and those from a gene body with a nucleotide length that is a multiple of three. Genomic GC values were calculated from the whole genome sequence. GC3 values are unweighted means of per gene GC3 value.

Our single-celled eukaryotic set were downloaded and filtered much in the same way. 70 eukaryote genomes of unique genus were downloaded from the full Ensembl Protist set (https://protists.ensembl.org/species.html, last accessed 8th August 2019). Similar to the ciliates analysis by Adachi and Cavalcanti [40], we extracted a sequence inclusive of the primary stop followed by 97 nucleotides of the 3' UTR from each gene. As with the bacterial genomes, we do this by applying annotated coding sequence coordinates to the total genomic sequence. Only genes with 3' intergenic space of >100 base pairs were considered for analysis to ensure a sample of genes with sufficient 3' UTR length. Extracted 3' UTR sequences were subjected to the same filters as with the bacterial ones. We increased our sample with the addition of two yeast species via bespoke downloads—*S. cerevisiae* (yeastgenome.org) and *C. albicans* (candidagenome.org). For *S. cerevisiae*, annotated 3' UTR coordinates were applied to the whole genome sequence to extract the appropriate sequence. For *C. albicans*, 3' UTR sequences were located downstream from the first in-frame stop codon of downloadable ORFs (that contain

intergenic sequence). We exclude genomes with < 500 qualifying 3' UTR sequences, leaving a final sample of 68 genomes.

## Protein abundance data

Experimental protein abundance data were downloaded for all genomes available from PaxDb [71]. Corresponding whole genome sequence files were downloaded from the European Molecular Biology Laboratory (EMBL) database. PaxDb external IDs and EMBL locus tags were extracted and matched to generate a sample of genomes and genes for which both PaxDb and EMBL sequence data were available (n = 24). In these genomes, qualifying genes that feature in the top and bottom quartiles of PaxDb data were defined as highly expressed genes (HEGs) and lowly expressed genes (LEGs) respectively. Only genomes with >100 qualifying HEGs and >100 qualifying LEGs were considered (n = 22). In reporting our results, we refer to the analysis of three gene groups: HEGs and LEGs which contain the qualifying genes of the 22 genomes for which there was available gene expression data, and 'all genes' where the qualifying genes of all filtered genomes are considered regardless of expression level.

## Simulations

ASC frequencies for codon positions +1 to +6 were compared to expected frequencies generated from a null model where sequence is dictated solely by 3' UTR dinucleotide content. To achieve this, we simulated 10,000 UTR sequences for each genome using Markov models to preserve reading frame context at the dinucleotide level. ASCs are likely to occur by chance in every genome at a given rate that is dependent on its dinucleotide content. Hence the observation of ASC frequencies that exceed our null represents enrichment beyond chance. Nucleotide frequencies used in the Markov decision process were determined by generating a string containing the 3' UTRs of all qualifying genes from a given genome. The raw frequencies of each nucleotide within this string were calculated for the selection of the first base of each simulation. Overlapping dinucleotide frequencies were calculated for the selection of following simulated nucleotides according to the previously selected nucleotide. Simulations were complete once 21 nucleotides in length (equivalent to a primary stop followed by 6 downstream codons).

For each genome, ASC frequencies were calculated and compared to the mean ASC frequencies from the 10,000 simulated sequences at each of the 6 downstream codon positions. Comparison to null was established through the calculation of Z-scores under the assumption of a normal distribution to assess the magnitude of deviation from null in standard deviation units. Z-scores were used to complete various binomial tests using the binom_test function from the SciPy stats R package [72].

## Translation table 4 mollicute analyses

The mollicute group contains both TT11 and TT4 genomes, allowing a side-by-side comparison in closely related species. TGA is not used as a stop codon in TT4 genomes. Hence, if observed TGA frequency is lower in TT4 genomes than in TT11 genomes, this implies selection upon TGA as an ASC in TT11 genomes. We design two tests to investigate whether TGA is underused in TT4 genomes.

(i) Frequency of TGA at codon positions +1 to +6 was plotted against genomic GC3 content in TT11 genomes from the full genome set (n = 644). A LOESS model was fit to allow the prediction of TGA frequency of TT11 and TT4 mollicute genomes according to their GC3 content at each position. TGA frequencies at each position for mollicute genomes were calculated

and compared to their predicted values. The fail-safe hypothesis predicts under enrichment of TGA in the TT4 genomes, but not TT11 ones.

(ii) Frequencies of TGA at positions +1 to +6 were calculated for TT4 mollicute genomes and compared to those of GC3 content-matched TT11 genomes from the full genome set. TT11 genomes were selected for comparison if their genomic GC3 content lies within 3.5% of the focal TT4 genome. Mean TGA frequencies for each position were calculated for selected TT11 genomes and compared with the corresponding TT4 genome frequency.

## Third stop frequency as a function of presence/absence of a prior ASC

Genes with an ASC were compared to those without. The null expectation is that those containing an ASC before (and including) position +N have an equal chance of possessing another ASC downstream as genes without one. Two groups of genes were thus extracted for each position–those with an ASC up to position N and those without. ASC frequencies of each group were calculated for downstream positions up to position +6 and compared. Given the nature of this experiment, no data is available for position +6 (as there is no further downstream position to use to calculate ASCs within our chosen intergenic range). To consider more localised nucleotide preferences, we also repeat this methodology considering just the following base (+N+1) instead of all downstream positions.

## Nucleotide enrichment at fourth and fifth nucleotide sites

For the analysis of the fourth nucleotide site of the primary stop codon, raw nucleotide frequencies (A, T, G, C) were calculated. Fourth site T enrichment relative to null was investigated through the comparison of T-starting codon frequency at position +1 to the mean frequency of T-starting codons throughout the 3' UTR (+1 to +6) using a Wilcoxon signed-rank test.

The analysis of the fifth nucleotide site of +4T-containing genes was completed in a similar manner. Raw nucleotide frequencies at nucleotide position +5 of genes were calculated, plotted for visual comparison and used in the completion of statistical analysis. Fifth site T and fifth site C enrichment relative to null was investigated through the comparison of TT/TC-starting codon frequency at position +1 to the mean frequency of TT/TC-starting codons throughout the 3' UTR (+1 to +6) of the given genome using a Wilcoxon signed-rank test.

## 3' stop codon switch analysis

Analysis of stop codon switches (from non-stop to stop, or vice versa) was completed by adapting a methodology described in previous studies [12, 73, 74]. Orthologous gene information for closely related species were downloaded from the Alignable Tight Genome Clusters (ATGC) database [75]. Corresponding whole genome sequence data was downloaded from NCBI [76]. Where possible, the same triplets (containing two closely related ingroup species and one outgroup to allow the reconstruction of mutations by a parsimony approach) were downloaded as used in previous studies [12, 73, 74]. In total, 29 ATGC triplet clusters were considered in the analysis (8 of the 37 clusters used in prior studies were ineligible).

All gene sequences from each ATGC-COG (Cluster of Orthologous genes) were aligned using the -linsi parameter of MAFFT [77]. Aligned genes without gaps downstream of the primary stop, from all genomes, were considered together in the codon switch analysis. Ancestral codons were inferred where the outgroup codon matched at least one of the ingroup codons. A switch was recorded where one of the ingroup codons differed from both the other ingroup codon and the outgroup codon (and thus the inferred ancestral codon). Frequencies of switches from non-stop to stop and stop to non-stop amongst 'in-frame' 3' UTR codons were

calculated. These were compared to null frequencies, calculated through the analysis of the same sequences but with +1 frameshift.

## Bacteria and single-celled eukaryote comparisons

To provide fair comparison between prokaryotes and eukaryotes we adopt the same set of methodologies for both genome sets. Due to the nature of ASC enrichment in eukaryotes not being universally specific to a particular codon position, we count the number of genomes in each set that possess ASC enrichment at one or more site (between +2 to +6). ASCs at a particular position were considered to be enriched if they produced a positive $Chi^2$ value and a p-value below 0.05/5 (after Bonferroni correction) when compared to the mean from a dinucleotide controlled null (see 'Simulations' section of these methods). As we set our p-value threshold at 0.01, the probability of a genome possessing significant ASC enrichment at one or more positions by chance is $0.99^5$ (approximately 0.951). Therefore, there is a $1-0.99^5$ (approximately 0.049) probability that a genome will contain significant enrichment at one or more positions by chance. We determined whether the number of genomes containing enrichment in each set was higher, lower, or as expected by using binomial tests under the null expectation that 4.9% of genomes possess enrichment purely by chance.

We additionally repeat the analysis using the null model proposed by Adachi and Cavalcanti [40]. In their analysis of ASC enrichment in ciliates, they consider the probability of finding the first in-frame stop codon as a function of 3' distance from the primary stop. The probability of finding the first stop at position +1 is equal to the probability of finding a stop at any position, *p*. The probability *p* is calculated for each genome by concatenating the first 100 non-coding nucleotides downstream of each gene, scanning this sequence for in-frame stops, and dividing the total number of stops by the total number of codon positions considered. At position +2, the probability of finding the first stop is the probability of not finding a stop at any position upstream, in this case position +1, multiplied by the probability of finding a stop at any position. This concept is recursively applied with each position downstream such that first ASC probability = $p[1-p]^{(n-1)}$, where n is the focal codon position. For each position +1 to +6 we calculate ASC probability and multiply this by the total number of UTR sequences analysed to determine the expected number. We then apply a $Chi^2$ test. To determine whether the number of genomes showing significant enrichment at one of more sites is higher, lower, or as expected, we apply a binomial test as described above.

## Supporting information

**S1 Fig. Z-scores measuring deviation from a null model (10,000 simulations) plotted against the genomic GC3 content of filtered TT11 bacterial genomes.** Significant negative relationships were observed between Z-score and genomic GC3 content at each position (Spearman's rank: $p < 2.2 \times 10^{-16}$ for all positions; $\rho = -0.61$ at position +1, $\rho = -0.65$ at position +2, $\rho = -0.51$ at position +3, $\rho = -0.45$ at position +4, $\rho = -0.41$ at position +5, $\rho = -0.46$ at position +6).
(TIF)

**S2 Fig. Gradient analysis of Z-score plotted against genomic GC3.** Raw gradients (of Z-score plotted against genomic GC3) plotted at each of the six positions downstream (**A**). Absolute gradients plotted against codon position (**B**). Our expectation under the fail-safe hypothesis is that at codon position +1, stops will be largely resistant to GC pressure while at position +6 this resilience will be diminished. We thus predict that looking across genomes, the plot of ASC usage against GC content should be flatter at site +1 than at site +6. Interestingly, there is

a significant correlation between absolute gradient and distance from the primary stop (Spearman's rank: p = 2.8 x $10^{-3}$ ρ = -1). We therefore infer that either the presence of ASCs is more resilient to GC pressure when located further downstream relative to the primary stop, or ASCs are actively selected against at positions closest to the primary stop. Both of these inferences go against the fail-safe hypothesis.
(TIF)

**S3 Fig. Raw ASC frequencies at positions +1 to +6 plotted against genomic GC3 content.** There is a significant negative correlation between the variables at all positions tested (Spearman's rank: p < 2.2 x $10^{-16}$ for all positions; ρ = -0.92 for position +1, ρ = -0.95 for position +2, ρ = -0.95 for position +3, ρ = -0.94 for position +4, ρ = -0.93 for position +5, ρ = -0.94 for position +6). The gradient of the linear model fitted for position +1 is significantly different than that of position +6 (p = 2.035732 x $10^{-10}$), with position +6 having the more negative gradient.
(TIF)

**S4 Fig. Gradient frequencies of HEGs compared to LEGs.** Each bar represents one genome, with genomes ordered by genomic GC3 content from left to right. Bar heights represents the raw difference between in ASC frequency between the two groups tested. A positive difference represents enrichment in the HEGs group, a negative difference represents enrichment in the LEGs group. There is a no significant difference between HEGs and LEGs at any position (Wilcoxon signed-rank test: p > 0.05/6).
(TIF)

**S5 Fig. Z-scores, measuring deviation of observed ASC frequencies in +4T-containing genes from +4T-containing null simulations, plotted against genomic GC3 content.** Only position +1 and position +2 are considered as these are the only sites where a signal for ASC enrichment has been noted. We find Z-scores to be negatively correlated with genomic GC3 when considering all genes at position +1 (Spearman's rank: ρ = -0.3054869, p < 2.2 x $10^{-16}$) and position +2 (Spearman's rank: ρ = -0.1880088, p = 1.62 x $10^{-06}$). There is no relationship between Z-score and genomic GC3 in HEGs or LEGs at either position (Spearman's rank, p > 0.05).
(TIF)

**S6 Fig. Assessment of fifth site nucleotide preferences.** Fifth site nucleotide frequencies in +4T-containing genes of different primary stop and expression level (**A**). Frequencies of TC (**B**) and TT-starting (**C**) codons at position +1 compared to the average frequency of the respective codons between positions +1 to +6. Positive scores represent enrichment whilst negative scores represent under-representation.
(TIF)

**S7 Fig. ASC frequencies of TT4 mollicute genomes calculated and compared to those of GC-matched TT11 genomes.** Each bar represents the frequency difference between a mollicute genome and the average of its GC-matched TT11 genomes. TGA was underrepresented at positions +3 and +5 only (Wilcoxon signed-rank tests: p = 0.11 for position +1; p = 0.15 for position +2; p = 1.5 x $10^{-3}$ for position +3; p = 0.70 for position +4; p = 6.8 x $10^{-4}$ for position +5; p = 0.11 for position +6).
(TIF)

**S8 Fig. Relative usage of each stop codon as the primary stop (position +0) and as ASCs at positions +1 to +6.** Contra to our expectations, we find codon usage at positions +1 to +6 to

be consistent with that of the primary stop.
(TIF)

**S1 Table. Positional codon switch (from stop to non-stop and non-stop to stop) counts and frequencies compared between the in-frame and out-of-frame 3' UTR codons of 29 triplets of closely related bacterial genomes.**
(TIF)

**S2 Table. Codon switch (from stop to non-stop and non-stop to stop) counts and frequencies compared between the in-frame and out-of-frame 3' UTR codons of a triplet of TT4 mollicute genomes.**
(TIF)

**S3 Table. 3' UTR frequencies of all codons in TT4 mollicutes compared to prediction by LOESS model fitted to TT11 codon frequencies.** Observed frequencies were compared to predicted frequencies using one-tailed Wilcoxon-signed rank tests, the p-values from which are found in the table below. A significant p-value represents significant under-enrichment in the TT4 genomes. Stop codons are highlighted in blue.
(TIF)

**S4 Table. Association between relative stop codon usage and GC3 content assessed by Spearman's Rank tests at each downstream position.**
(TIF)

**S5 Table. Association between relative stop codon usage and GC3 content assessed by Spearman's Rank tests in each reading frame.**
(TIF)

**S6 Table. Association between gene length and gene expression level assessed at genome-wide level.** Gene expression (represented by experimental protein abundance data) and gene nucleotide lengths were used in Spearman's rank tests.
(TIF)

**S7 Table. Z-score analysis of single-celled eukaryotic genomes.** Z-scores represent deviation in ASC frequency from dinucleotide-controlled simulations.
(TIF)

**S8 Table. Chi$^2$ analysis of ASC frequency single-celled eukaryotic genomes against dinucleotide-controlled simulations.**
(TIF)

**S9 Table. Chi$^2$ analysis of ASC frequency single-celled eukaryotic genomes against the Adachi and Cavalcanti null.**
(TIF)

**S1 Text. Supporting text for S1 Fig, S2 Fig and S3 Fig.**
(DOCX)

**S2 Text. Supporting text for S1 Table and S2 Table.**
(DOCX)

**S3 Text. Supporting text for S4 Fig.**
(DOCX)

**S4 Text. Supporting text for S5 Fig.**
(DOCX)

**S5 Text. Supporting text for S6 Fig.**
(DOCX)

**S6 Text. Supporting text for S7 Fig.**
(DOCX)

**S7 Text. Supporting text for S3 Table.**
(DOCX)

**S8 Text. Supporting text for S8 Fig, S4 Table and S5 Table.**
(DOCX)

**S9 Text. Supporting text for S6 Table.**
(DOCX)

**S10 Text. Supporting text for S7 Table, S8 Table and S9 Table.**
(DOCX)

## Author Contributions

**Conceptualization:** Alexander T. Ho, Laurence D. Hurst.

**Data curation:** Alexander T. Ho.

**Formal analysis:** Alexander T. Ho, Laurence D. Hurst.

**Funding acquisition:** Laurence D. Hurst.

**Investigation:** Alexander T. Ho.

**Methodology:** Alexander T. Ho, Laurence D. Hurst.

**Project administration:** Alexander T. Ho, Laurence D. Hurst.

**Supervision:** Laurence D. Hurst.

**Visualization:** Alexander T. Ho.

**Writing – original draft:** Alexander T. Ho.

**Writing – review & editing:** Alexander T. Ho, Laurence D. Hurst.

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
