## [Decision Letter · Decision Letter 0]

25 Jul 2019

Dear Dr Ho,

Thank you very much for submitting your Research Article entitled 'In eubacteria, unlike eukaryotes, there is no evidence for selection favouring fail-safe 3’ additional stop codons' to PLOS Genetics. Your manuscript was fully evaluated at the editorial level and by three independent peer reviewers. The reviewers appreciated the attention to an important problem, but raised some substantial concerns about the current manuscript. Based on the reviews, we will not be able to accept this version of the manuscript, but we would be willing to review again a much-revised version. We cannot, of course, promise publication at that time.

If you decide to revise the manuscript for further consideration at PLOS Genetics, please aim to resubmit within the next 60 days, unless it will take extra time to address the concerns of the reviewers, in which case we would appreciate an expected resubmission date by email to plosgenetics@plos.org.

[LINK]

We are sorry that we cannot be more positive about your manuscript at this stage. Please do not hesitate to contact us if you have any concerns or questions.

Yours sincerely,

Xavier Didelot

Associate Editor

PLOS Genetics

Kirsten Bomblies

Section Editor: Evolution

PLOS Genetics

Reviewer's Responses to Questions

**Comments to the Authors:**

Reviewer #1: This is an interesting study demonstrating that eubacteria do not seem to contain additional stop codons down-stream from the primary stop codon of coding sequences. This pattern is in contrast to patterns observed in certain eukaryotes, whose 3' sequences downstream of stop codons are enriched in further stop codons.

The manuscript is carefully done and comprehensive. I have no concerns about the data analysis. However, I do have one substantive comment. In yeast, I'm aware that stop codon read through can be regulated via the yeast prion [PSI+] (see e.g., your Ref. 6 (Drummond and Wilke 2019), Fig. 3B, and references therein). I'm not sure whether a similar mechanism exists in ciliates. In any case, the [PSI+] state leads to extensive read through, much more than is observed under normal physiological conditions, and I could imagine that it would require extensive additional stop codons to be managed. If no such mechanism exists in eubacteria, then the selection pressure for 3' downstream stop codons would be much reduced.

I don't expect the authors to resolve this issue in their manuscript, but I think it would be warranted to add a paragraph or two about [PSI+] to the discussion. Currently, this mechanism isn't mentioned at all, and that seems like a major omission to me.

Minor comment:

- Throughout, please verify that all figure elements are sized appropriately so they are legible. Specifically, the legends in Fig. 7 are definitely too small.

Reviewer #2: This paper is well-written, except that it lacks scientific rigor both in building up the case and in drawing conclusions. I will give an example in each.

In building up the case, the authors stated that “adenine enrichment at the fourth coding sequence residue in bacterial genes promotes translation termination following a frameshift event at the initiating ATG that allows an out-of-frame stop codon to be read (9,10)”. I expect to see experimental evidence, but there isn’t any. The two papers cited in support of the claim are quite speculative. Two groups of scientists saying that something is possible does not equal to make that something an actuality.

In making conclusions, the author stated that “Contra to the predictions of the hypothesis we find: there is paucity, not enrichment, of ASCs downstream”. This is not correct. I retrieved some compilation I did a long time ago on Escherichia coli that recorded the additional stop codon (ASC) occurrence downstream of the annotated codon (24 bases immediately downstream of the annotated codon). A stop codon is much more likely to occur immediately following the annotated codon than several triplets away. There is a highly significant negative correlation between the presence of a stop codon and the distance downstream from the annotated codon. I have verified this with the most recent E. coli genome. This finding is indeed what one would expect from the fail-safe mechanism. If fail-safe mechanism is to work well, it is better to have it right after the annotated stop codon than many based away downstream from it.

I could offer more examples, but I believe that this is sufficient for the authors.

Reviewer #3: The authors have addressed here a very important question i.e. whether there exists ‘additional stop codons (ASC)’ as a ‘fail-safe’ mechanism for ‘stop codon read-through’ in eubacteria. Contradictory to the common belief they find that there is no enrichment (but, rather lack) of ASCs downstream, the correlation of ASC frequency is negative in relation to GC content of the genome.

While the work is highly interesting, I have a major criticism i.e. the lack of proper control for the method they have used. I elaborate the concern here. The authors have investigated the occurrence of ASCs by comparing with a dinucleotide controlled null. The logic behind this is not very clear. Since the whole conclusion is dependent on this primary analysis, the authors must provide results with positive and negative controls. It is well established that the eukaryotic genomes contain ASCs downstream the genes. The authors must validate their method by estimating Z score using the same method with equivalent number of eukaryotic genomes. Then the additional results can be accepted.

I also wonder why the analysis is restricted to only 600 genomes, while more than 5000 eubacterial genome sequences are available.

There are mistakes in formatting the in-text references (e.g. page 19 - (Major, et al. 2002; Korkmaz, et al. 2014; Wei and Xia 2017)), which should be corrected.

**Have all data underlying the figures and results presented in the manuscript been provided?**

Reviewer #1: Yes

Reviewer #2: Yes

Reviewer #3: None

PLOS authors have the option to publish the peer review history of their article (what does this mean?). If published, this will include your full peer review and any attached files.

Reviewer #1: No

Reviewer #2: No

Reviewer #3: No

---

## [Editor Report · Decision Letter 1]

27 Aug 2019

Dear Dr Ho,

We are pleased to inform you that your manuscript entitled "In eubacteria, unlike eukaryotes, there is no evidence for selection favouring fail-safe 3’ additional stop codons" has been editorially accepted for publication in PLOS Genetics. Congratulations!

Yours sincerely,

Xavier Didelot

Associate Editor

PLOS Genetics

Kirsten Bomblies

Section Editor: Evolution

PLOS Genetics

**Data Deposition**

http://datadryad.org/submit?journalID=pgenetics&manu=PGENETICS-D-19-00941R1

**Press Queries**

---

## [Editor Report · Acceptance letter]

13 Sep 2019

PGENETICS-D-19-00941R1 

In eubacteria, unlike eukaryotes, there is no evidence for selection favouring fail-safe 3’ additional stop codons 

Dear Dr Ho, 

We are pleased to inform you that your manuscript entitled "In eubacteria, unlike eukaryotes, there is no evidence for selection favouring fail-safe 3’ additional stop codons" has been formally accepted for publication in PLOS Genetics! Your manuscript is now with our production department and you will be notified of the publication date in due course.

With kind regards,

Kaitlin Butler

PLOS Genetics

On behalf of:
